# Continuous sensing of IFNα by hepatic endothelial cells shapes a vascular antimetastatic barrier

**Ngoc Lan Tran[1†‡], Lorena Maria Ferreira[1†], Blanca Alvarez-Moya[1†], Valentina Buttiglione[1§], Barbara Ferrini[1], Paola Zordan[1,2], Andrea Monestiroli[1], Claudio Fagioli[1], Eugenia Bezzecchi[3], Giulia Maria Scotti[3], Antonio Esposito[2,4], Riccardo Leone[2,4], Chiara Gnasso[2,4], Andrea Brendolan[5], Luca G Guidotti[1,2], Giovanni Sitia[1]\***

[1]Division of Immunology, Transplantation and Infectious Diseases, IRCCS San Raffaele Scientific Institute, Milan, Italy; [2]Vita-Salute San Raffaele University, Milan, Italy; [3]Center for Omics Sciences, IRCCS San Raffaele Hospital, Milan, Italy; [4]Experimental Imaging Center, IRCCS San Raffaele Scientific Institute, Milan, Italy; [5]Division of Experimental Oncology, IRCCS San Raffaele Scientific Institute, Milan, Italy

**\*For correspondence:**
sitia.giovanni@hsr.it

[†]These authors contributed equally to this work

**Present address:** [‡]University of Geneva, Geneva, Switzerland; [§]Ipsen Spa, Assago, Italy

**Abstract** Hepatic metastases are a poor prognostic factor of colorectal carcinoma (CRC) and new strategies to reduce the risk of liver CRC colonization are highly needed. Herein, we used mouse models of hepatic metastatization to demonstrate that the continuous infusion of therapeutic doses of interferon-alpha (IFNα) controls CRC invasion by acting on hepatic endothelial cells (HECs). Mechanistically, IFNα promoted the development of a vascular antimetastatic niche characterized by liver sinusoidal endothelial cells (LSECs) defenestration extracellular matrix and glycocalyx deposition, thus strengthening the liver vascular barrier impairing CRC trans-sinusoidal migration, without requiring a direct action on tumor cells, hepatic stellate cells, hepatocytes, or liver dendritic cells (DCs), Kupffer cells (KCs) and liver capsular macrophages (LCMs). Moreover, IFNα endowed LSECs with efficient cross-priming potential that, along with the early intravascular tumor burden reduction, supported the generation of antitumor CD8+ T cells and ultimately led to the establishment of a protective long-term memory T cell response. These findings provide a rationale for the use of continuous IFNα therapy in perioperative settings to reduce CRC metastatic spreading to the liver.

## Editor's evaluation

This study describing how continuous perioperative IFNα therapy stimulates hepatic endothelial cells to build up a physical vascular barrier that limits tumor cell entry into the liver and promotes long-term antitumor immunity provides novel evidence for anti-metastatic effects of IFNα via effects on the liver vascular compartment. This work is predicted to advance the field and will be of interest to scientists studying cancer, inflammation and liver function.

## Introduction

Colorectal cancer (CRC) is the third most common cancer and the second leading cause of cancer-related death worldwide (*Sung et al., 2021*). Surgical resection of the primary CRC tumor is the mainstay of treatment (*Argilés et al., 2020*; *Cunningham et al., 2010*; *Seo et al., 2013*) unfortunately, up to 50% of these patients – despite chemotherapy and targeted adjuvant therapies – often develop life-threatening liver metastatic disease in the following years (*Argilés et al., 2020*; *Sargent et al.,*

**eLife digest** Colorectal cancer remains one of the most widespread and deadly cancers worldwide. Poor health outcomes are usually linked to diseased cells spreading from the intestine to create new tumors in the liver or other parts of the body. Treatment involves surgically removing the initial tumors in the bowel, but patient survival could be improved if, in parallel, their immune system was 'boosted' to destroy cancer cells before they can form other tumors.

Interferon alpha is a small protein which helps to coordinate how the immune system recognizes and deactivates foreign agents and cancerous cells. It has recently been trialed as a colorectal cancer treatment to prevent tumors from spreading to the liver, but only with limited success. This partly because interferon-alpha is usually administered in high and pulsed doses, which cause severe side effects through the body.

Instead, Tran, Ferreira, Alvarez-Moya et al. aimed to investigate whether continuously delivering lower amounts of the drug could be a better approach. This strategy was tested on mice in which colorectal cancer cells had been implanted into the wall of the large intestine. Continuous administration minimized the risk of the implanted cancer cells spreading to the liver while also creating fewer side effects. The team was able to identify an optimum delivery strategy by varying how much interferon-alpha the animals received and when.

Further experiments also revealed a new mechanism by which interferon-alpha prevented the spread of colorectal cancer. Upon receiving continuous doses of the drug, a group of liver cells started to generate a physical barrier which stopped cancer cells from being able to invade the organ. The treatment also promoted long-term immune responses that targeted diseased cells while being safe for healthy tissues. If confirmed in clinical trials, these results suggest that colorectal patients undergoing tumor removal surgery may benefit from also receiving interferon-alpha through continuous delivery.

2009). While the overall benefit of surgery is well established (*Seo et al., 2013*), it has been also proposed that this procedure may foster liver metastases by increasing the dissemination of CRC cells into the portal circulation (*Chow and Chok, 2019*; *Denève et al., 2013*), enhancing the adhesion of CRC cells to the liver endothelium (*Chambers et al., 2002*; *Gül et al., 2011*) or promoting transient immunosuppression awakening dormant intrahepatic micrometastases (*Ananth et al., 2016*).

Accordingly, there is growing recognition that the use of perioperative immunotherapies in CRC patients undergoing surgical resection may represent a unique treatment window to prevent metastatic colonization and control minimal residual disease (*Badia-Ramentol et al., 2021*; *Bakos et al., 2018*; *Horowitz et al., 2015*). In this context, interferon-alpha (IFNα), a pleiotropic cytokine with multiple antitumor effects such as the direct inhibition of cancer cell growth and angiogenesis (*Indraccolo, 2010*), the sustained upregulation of major histocompatibility complexes (*Gessani et al., 2014*) and the induction of innate and adaptive antitumor immune responses (*Aichele et al., 2006*; *Curtsinger et al., 2007*; *Fuertes et al., 2013*), has been used as adjuvant immunotherapy in various solid cancers such as renal cell carcinoma (*Flanigan et al., 2001*), melanoma (*Lens and Dawes, 2002*) and colorectal cancer (*Köhne et al., 1997*; *Link et al., 2005*). Unfortunately, systemic administration of IFNα has shown limited clinical efficacy, likely due to its short plasma half-life (~1 hr) (*Bocci, 1994*) and the use of high and pulsed doses, which often resulted in systemic side effects (*Weber et al., 2015*). To overcome these limitations, several strategies to prolong IFNα half-life and target the tumor microenvironment have been tested (*Fioravanti et al., 2011*; *Herndon et al., 2012*; *Jeon et al., 2013*; *Li et al., 2017*; *Liang et al., 2018*; *Yang et al., 2014*), including a preclinical gene/cell therapy approach that can deliver constant amounts of IFNα into the liver to significantly curb CRC metastatic growth (*Catarinella et al., 2016*).

Herein, we adopted a continuous intraperitoneal (ip) IFNα delivery strategy to show that steady and tolerable IFNα doses reduce liver CRC metastatic spreading and improves survival in several CRC mouse models. Our results showed that the antimetastatic effects of IFNα rely neither on the direct inhibition of tumor cell proliferation nor on the indirect stimulation of hepatocytes, hepatic stellate cells, liver DCs, Kupffer cells (KCs) and liver capsular macrophages (LCMs). Rather, the results identify HECs, including LSECs, as key mediators of IFNα-dependent anti-tumor activities that involve the

impairment of CRC trans-sinusoidal migration and the development of long-term anti-tumor CD8[+] T cell immunity.

## Results

### Selection of the optimal IFNα dosing regimen

To avoid well-known toxicities, especially myelotoxicity, caused by high IFNα doses (*Weber et al., 2015*) and to define a delivery strategy providing prolonged and non-fluctuating IFNα levels in blood and tissues, normal inbred mice were implanted intraperitoneally with mini-osmotic pumps (MOP) constantly releasing different rates (i.e. 50 ng/day, 150 ng/day, or 1050 ng/day) of recombinant mouse IFNα1 (termed IFNα from now on) over time. Serum IFNα levels peaked at day 2 after MOP implantation and relative IFNα amounts (from ~100 pg/ml to ~1200 pg/ml) reflected the different MOP loading doses (*Figure 1A*). Serum IFNα levels decreased, albeit not uniformly, at days 5 and 7 post implantation (*Figure 1A*), mirroring the pharmacokinetic-pharmacodynamic (PK-PD) behavior of other long-lasting formulations of IFNα (*Jeon et al., 2013*). A reduction in circulating white blood cells (WBCs) but not in platelet (PLT) counts or hematocrit (HCT) was detected only with the highest dose (*Figure 1B* and *Figure 1—figure supplement 1A,B*). Looking at liver toxicity, we observed no increases in serum alanine aminotransferase (sALT) at all tested doses and time points (*Figure 1—figure supplement 1C*) and no abnormal changes in liver morphology at autopsy (*Figure 1—figure supplement 1D*). Looking at the intrahepatic induction of the interferon-stimulated gene (ISG) *Irf7* (*Cheon et al., 2014*) at day 7, we observed a proportional dose response, with a six-fold increase in *Irf7* expression at the 150 ng/day dosing regimen (*Figure 1C*). Notably, this increase paralleled the increase that we previously documented to be associated with protection against liver CRC colonization following a gene/cell therapy based IFNα delivery strategy (*Catarinella et al., 2016*). Lack of bone marrow and liver toxicity, proper induction of hepatic *Irf7* expression and maintained responsiveness of hepatic liver cells to IFNα (*Figure 1—figure supplement 1E*) prompted us to select the 150 ng/day dosing regimen for follow-up investigations.

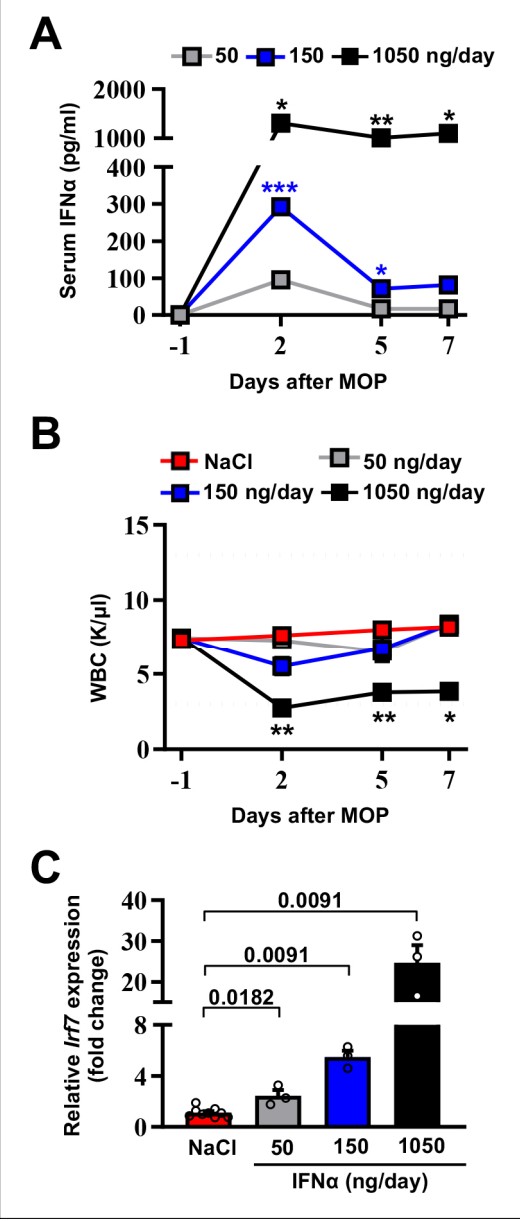

**Figure 1.** Selection of the optimal IFNα dosing regimen. (**A**) Quantification of plasma IFNα concentration from mice continuously treated with different IFNα doses at the indicated time points. Mean values ± SEM are shown; p-values were calculated by one-way ANOVA Tukey's multiple comparison test. Significant p-values refer to the IFNα 50 ng/day group, since NaCl-treated animals had IFNα plasma levels below the assay detection limit. p≤0.05; **p≤0.01; ***p≤0.001. (**B**) White blood cell (WBC) counts of mice treated with different IFNα doses at indicated time points. Horizontal dashed lines delimit normal WBC range. Mean value ± SEM are shown; p-values were calculated by one-way ANOVA Tukey's multiple comparison test. Significant p-values are referred to the NaCl group. *p≤0.05; **p≤0.01. (**C**) Quantitative real-time PCR analysis of *Irf7* mRNA expression from

*Figure 1 continued on next page*

*Figure 1 continued*

liver tissues of mice treated with different IFNα doses for 7 days. Basal *Irf7* mRNA expression level was determined in control mice (NaCl; n=9) and the relative expression of *Irf7* upon IFNα treatment is shown (n=3, for each IFNα-treated group). Mean values ± SEM are shown; p-values were calculated by Mann-Whitney test.

The online version of this article includes the following source data and figure supplement(s) for figure 1:

**Source data 1.** Data for the dose-response curves in the figure.

**Figure supplement 1.** Selection of the optimal IFNα dosing regimen.

**Figure supplement 1—source data 1.** Data for the dose-response curves in the figure.

## Continuous IFNα administration reduces liver CRC metastatic burden and improves survival

We next tested the ability of continuous IFNα administration (150 ng/day for 28 days) to reduce CRC metastatic growth in the liver. Groups of H-2$^{bxd}$ F1 hybrids of C57BL/6 J x BALB/c (CB6) mice were implanted with either control MOP-NaCl (termed NaCl) or MOP-IFNα (termed IFNα) (*Figure 2A*). Seven days later, a time frame compatible with the perioperative period in humans (*Horowitz et al., 2015*), CB6 mice were intrasplenically challenged with either the immunogenic microsatellite instable (MSI) MC38 CRC cell line (*Efremova et al., 2018*; *Rosenberg et al., 1986*) or the poorly immunogenic microsatellite stable (MSS) CT26 CRC cell line (*Brattain et al., 1980*; *Efremova et al., 2018*). Rapid removal of the spleen after CRC cell injection was implemented to avoid intrasplenic tumor growth (*Catarinella et al., 2016*). Each CRC cell line was injected at doses known to induce similar survival rates in age- and sex-matched CB6 recipients that, carrying hybrid H-2$^{bxd}$ alleles, are immunologically permissive to both MC38 and CT26 cells (*Catarinella et al., 2016*). After treatment initiation, well-tolerated serum IFNα levels of ~300 pg/ml at day 2 and ~100 pg/ml thereafter were observed (*Figure 2B* and *Figure 2—figure supplement 1A,B*), which subsequently declined to undetectable levels. The intrahepatic expression of *Irf7* monitored at day 21 after continuous IFNα therapy (*Figure 2—figure supplement 1C, D*) was like that observed earlier at day 7 (*Figure 1C*). Magnetic resonance imaging (MRI)-based longitudinal analyses – in MC38- (*Figure 2C* and *Figure 2—figure supplement 1E*) or CT26-challenged (*Figure 2D* and *Figure 2—figure supplement 1F*) animals – revealed that 100% of NaCl-treated mice (*Figure 2E and F*) develop multiple metastatic tumor lesions by days 21 and 28 after challenge and no mice displayed detectable tumors in other organs. Liver lesions increased in volume afterwards, ultimately resulting in imposed humane euthanization of both MC38 (*Figure 2—figure supplement 1I*) or CT26 (*Figure 2—figure supplement 1J*) tumor carriers in the intervening weeks. Conversely, 45% and 66% of IFNα-treated mice challenged with MC38 or CT26 cells, respectively, showed absence of liver metastases throughout the entire duration of the experiment (*Figure 2E and F*). All the remaining mice scoring disease-positive at days 21 and 28 displayed lesions that were reduced in number and size when compared to those detected in NaCl-treated counterparts (*Figure 2—figure supplement 1G, J*). Of note, metastatic lesions eventually regressed and achieved complete remission by day 50 in approximately 33% of IFNα-treated mice that were challenged with MC38 cells and scored disease-positive at day 21 (*Figure 2C and E* and *Figure 2—figure supplement 1E*), whereas none of the few CT26-challenged mice scoring disease-positive at day 21 survived long-term (*Figure 2D and F* and *Figure 2—figure supplement 1F*), with tumors confined only to the liver. Continuous IFNα administration also improved survival, with similar rates for both MC38- and CT26-challenged mice (*Figure 2G and H*). These results indicate that the continuous IFNα administration safely and efficiently limits the liver metastatic colonization of CRC cell lines intrinsically carrying different immunogenic or genetic properties.

## Continuous IFNα administration prevents spontaneous hepatic colonization of orthotopically implanted CT26$^{LM3}$ cells

To confirm the above-mentioned results in a different metastatic setting, we developed an orthotopic CRC model of liver metastases by implanting invasive CRC cells into the mouse cecal wall. As previously reported (*Zhang et al., 2013*), invasive CRC cells were generated by serial intracecal injections of the parental CT26 cells into CB6 mice (*Figure 3—figure supplement 1A*). The percentage of metastatic livers in intracecally implanted mice significantly increased as CT26 cells were passaged, with an almost 100% of animals bearing multiple liver metastases after 3 rounds of in vivo selection

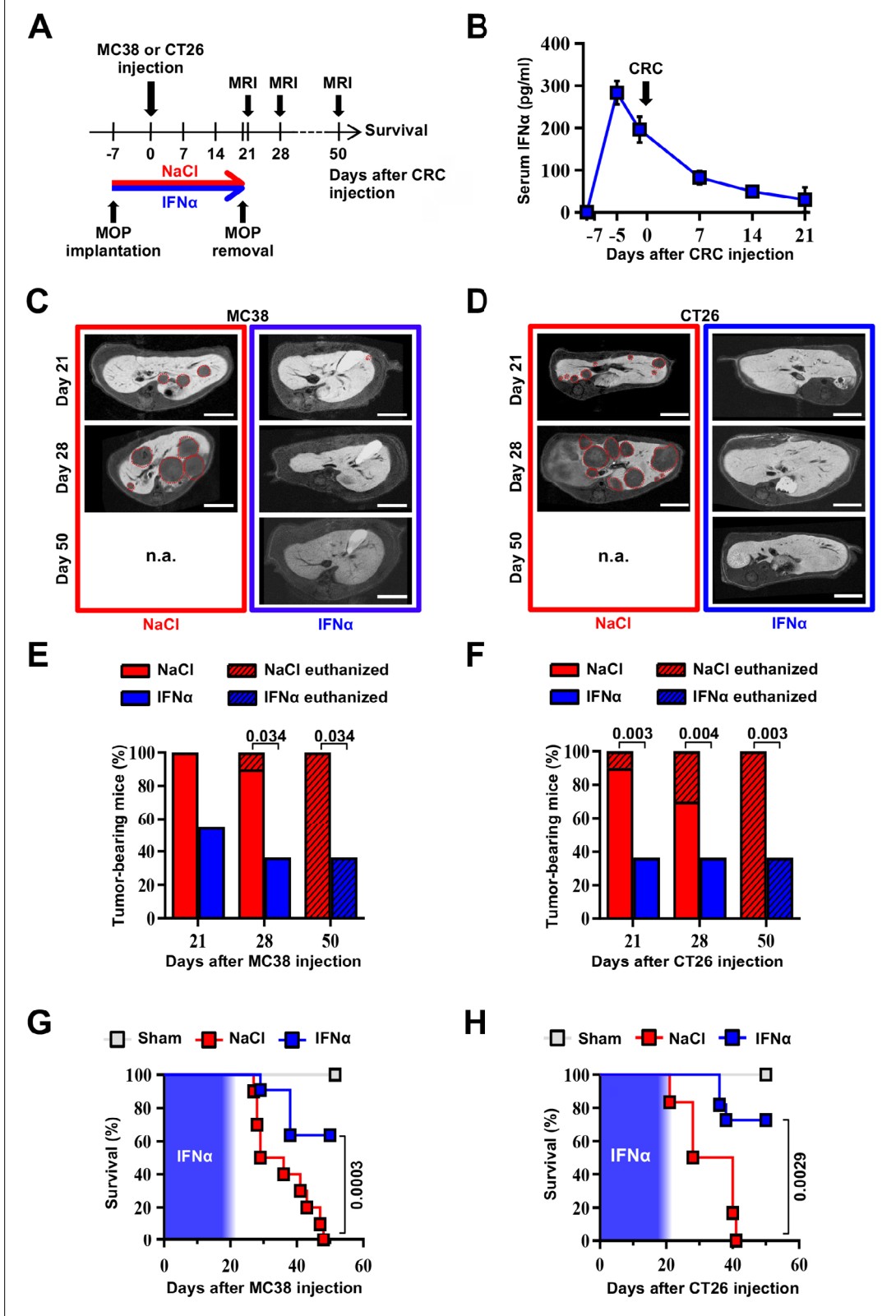

**Figure 2.** Continuous IFNα administration reduces liver CRC metastatic burden and improves survival. (**A**) Schematic representation of the experimental procedure. Intrasplenic injection of $7\times10^4$ MC38 or $5\times10^3$ CT26 cells was performed 7 days after continuous NaCl or IFNα therapy. (**B**) Quantification of plasma IFNα concentration at different time points after continuous IFNα administration (n=6). The time of intrasplenic CRC cell injection has also been depicted. Mean values ± SEM are shown. (**C–D**) Representative T1 contrast-enhanced magnetic resonance images (MRI) of the liver of mice treated

*Figure 2 continued on next page*

*Figure 2 continued*

with NaCl (red frame) and IFNα (blue frame) at 21, 28, and 50 days after MC38 (**C**) or CT26 (**D**) cells injection. Red dashed lines highlight CRC liver metastases, characterized as hypointense regions in T1-weighted sequences. n.a.=not assessed, is referred to mice euthanized before the specified time point; scale bar = 5 mm. (**E–F**) Percentage of mice treated with NaCl (MC38 n=3 + 3; CT26 n=5 + 5 for each of two independent experiments) or IFNα (MC38 n=5 + 6; CT26 n=5 + 6 for each of two independent experiments) bearing at least one CRC liver metastasis estimated by MRI analysis at indicated time points after MC38 or CT26 injection. The oblique black line pattern within columns depicts the percentage of mice euthanized before the indicated time point. Mean values are shown; p-values were calculated by Fisher's exact test. (**G–H**) Kaplan-Meier survival curves of Sham (n=3), NaCl-(MC38 n=6; CT26 n=10) or IFNα-treated (MC38 n=11; CT26 n=11) mice after MC38 or CT26 cells injection. The blue pattern indicates the time frame of IFNα ip release; p-values were calculated by log-rank/Mantel-Cox test.

The online version of this article includes the following source data and figure supplement(s) for figure 2:

**Source data 1.** Data for the graphs in the figure.

**Figure supplement 1.** Continuous intraperitoneal IFNα administration reduces MC38 and CT26 metastatic tumor burden without causing significant hematologic toxicity.

**Figure supplement 1—source data 1.** Data for the graphs in the figure.

(*Figure 3—figure supplement 1B-D*). Three-time passaged cells (termed CT26^LM3) were then orthotopically implanted in the cecal wall of CB6 mice and 7 days later the animals were treated with either NaCl or IFNα (*Figure 3A*).

Consistent with our previous results (*Figure 2B*), serum IFNα levels peaked at day 2 after MOP implantation (*Figure 3B*), without causing myelotoxicity (*Figure 3C*), and MRI analyses performed 14 days later revealed that continuous IFNα therapy did not alter the growth of primary intracecal tumors (*Figure 3D and E*), while IFNα treatment significantly reduced both number and size of hepatic lesions (*Figure 3D and F*) with 60% of mice spared from metastatic lesions (*Figure 3H*). The primary intracecal tumors (*Figure 3—figure supplement 2A*) and liver metastases (*Figure 3G*) detected after orthotopic implantation of CT26^LM3 cells were also characterized by immunohistochemistry (IHC). This analysis showed that primary intracecal tumors and liver metastatic lesions of NaCl-treated control mice were highly proliferative (as denoted by Ki67 positivity), exhibited marked signs of angiogenesis (as denoted by CD34 staining) and, accordingly with previous reports (*Catarinella et al., 2016*; *Tauriello et al., 2018*), were devoted of F4/80^+ resident macrophages and CD3^+ T cells (*Figure 3G and H*). Similar results were also observed in IFNα-treated primary intracecal tumors (*Figure 3—figure supplement 2A, B*). The absence of liver metastases in the majority of IFNα-treated mice is reflected by a reduced Ki67 or CD34 staining and an apparently normal distribution of F4/80^+ macrophages and CD3^+ T cells (*Figure 3G and H*). The few small hepatic lesions detected in 40% of mice continuously treated with IFNα (*Figure 3H* and *Figure 3—figure supplement 2C, D*) did not show differences in Ki67 positivity, CD34 staining or amount of F4/80^+ resident macrophages and CD3^+ T cells in relation to NaCl-treated mice (*Figure 3—figure supplement 2C, D*), consistent with the notion that CRC tumors may deregulate the *Ifnar1* receptor and, thus, become refractory to IFNα therapy (*Boukhaled et al., 2021*; *Katlinski et al., 2017*).

Altogether, these results indicate that continuous IFNα therapy does not significantly alter the growth of primary established CRC tumors but reduces the liver metastatic potential of invasive CRC cells emerging from the cecum.

## HECs mediate the anti-metastatic activity of IFNα

As the *Ifnar1* surface receptor subunit is necessary to mediate the pleiotropic anti-tumor properties of IFNα (*Cheon et al., 2014*), we deleted this molecule from CRC cells and from hepatic parenchymal and non-parenchymal cells to identify the mechanism of action (MoA) of continuous IFNα administration in our in vivo system. We restricted these studies to MC38 CRC cells because the genetic background of the mouse models did not allow us to use CT26 CRC MHC-I mismatched cell lines in C57BL/6 recipients. We also adopted a new seeding approach that – involving the injection of CRC cells through the superior mesenteric vein – potentially avoids immune deregulations linked to the splenectomy procedure. Seven days after NaCl or IFNα administration, C57BL/6 mice were challenged with wild-type MC38 cells or with MC38 cells that were CRISPR-Cas9-edited to lack a functional *Ifnar1* receptor (MC38^Ifnar1_KO). To this end, MC38-edited clones showed mismatches in a T7E1 assay and clone C8 (MC38^C8) carrying *Ifnar1* deleting mutations failed to express *Irf7* upon in vitro IFNα stimulation (*Figure 4—figure supplement 1A-C*).

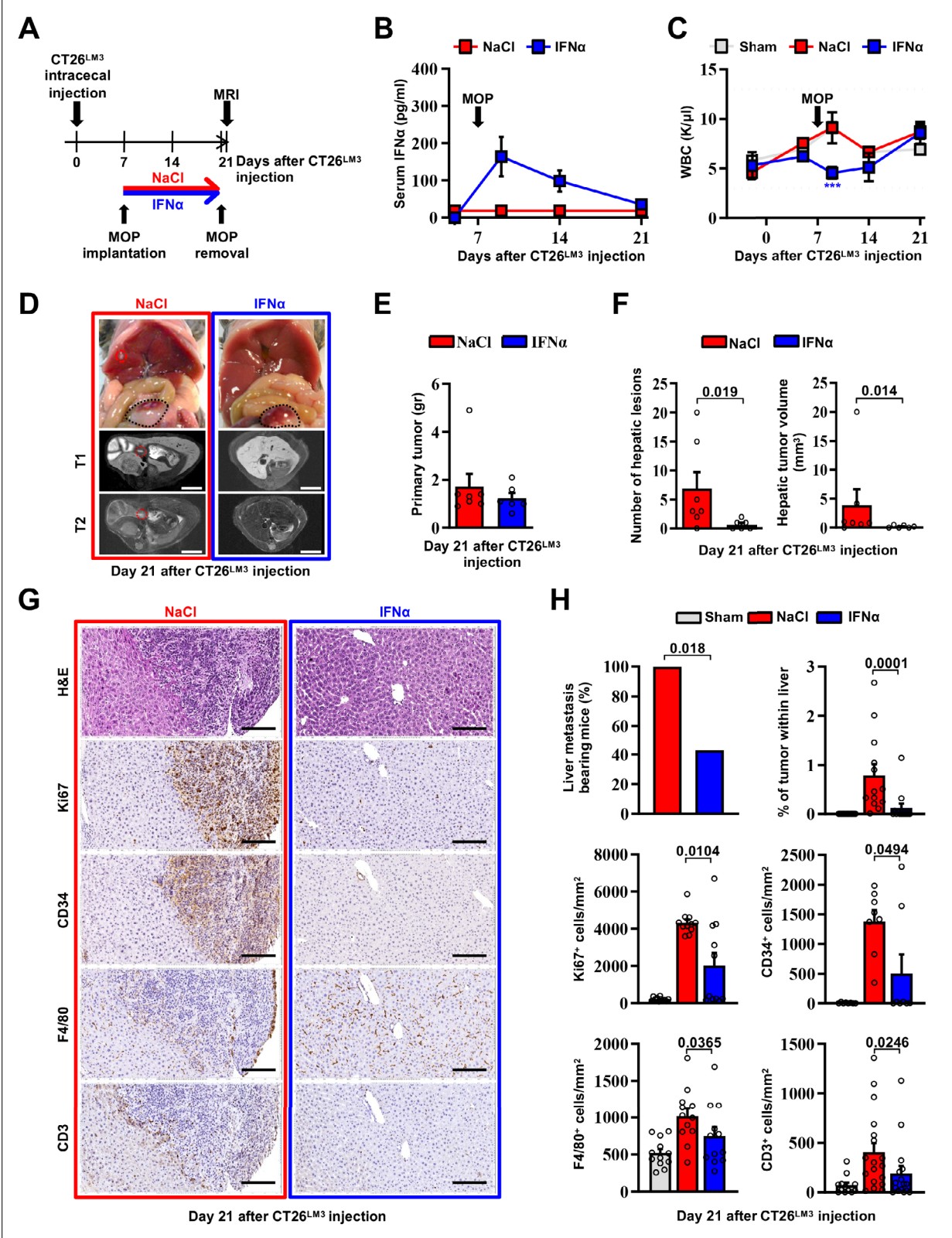

**Figure 3.** Continuous IFNα administration prevents spontaneous hepatic colonization of orthotopically implanted CT26$^{LM3}$ cells. (**A**) Schematic representation of the experimental procedure. Seven days after intracecal injection of 2x10$^5$ CT26$^{LM3}$ cells, mice were randomly assigned to receive either continuous NaCl (n=7) or IFNα (n=6) administration and analyzed by MRI 14 days later. (**B**) Quantification of plasma IFNα concentration at the indicated time point after cecal wall injection of CT26$^{LM3}$ cells in mice described in a. The arrow indicates the time of NaCl or IFNα therapy initiation.

*Figure 3 continued on next page*

*Figure 3 continued*

Mean values ± SEM are shown. (**C**) WBC counts from mice described in (**A**) continuously treated with NaCl or IFNα at indicated time points. Horizontal dashed lines delimit the normal WBC range. The time of MOP implantation has also been depicted. Mean value ± SEM are shown; p-values were calculated by one-way ANOVA Tukey's multiple comparison test. Significant p-values are referred to NaCl group. ***p≤0.001. (**D**) Representative images (top panels) of the hepatic lesions and intracecal tumors observed in NaCl- (red frame) and IFNα-treated (blue frame) mice, 21 days after CT26^LM3^ cells intracecal wall injection, and the corresponding hepatic contrast-enhanced MRI, T1-weighted (middle panels) and T2-weighted (bottom panels) sequences. Red dashed lines identify macroscopic liver metastatic lesions. Scale bars = 5 mm. (**E**) Quantification of the weight of primary CRC tumors 21 days after CT26^LM3^ cells intracecal wall injection of mice described in D. (**F**) Quantification of the number of hepatic lesions and total tumor volume of liver metastases by MRI analysis of mice described in D. Mean values ± SEM are shown; p-values were calculated by Mann-Whitney test. (**G**) Representative H&E, Ki67, CD34, F4/80, and CD3 immunohistochemical micrographs of liver metastatic lesions found in NaCl- (red frame) and IFNα-treated (blue frame) mice, 21 days after intracecal injection of CT26^LM3^ cells. Scale bar = 100 μm. (**H**) Quantification of the percentage of mice bearing liver metastases, as well as the percentage of tumor area and the number of cells expressing Ki67, CD34, F4/80, and CD3 per mm² determined by IHC. Immunohistochemical measurements were conducted on at least 1000 mm² of total liver area for both experimental conditions. Mean values ± SEM are shown; p-values were calculated by Mann-Whitney test.

The online version of this article includes the following source data and figure supplement(s) for figure 3:

**Source data 1.** Data for the graphs in the figure.

**Figure supplement 1.** In vivo selection of CT26^LM3^ cells with increased spontaneous liver metastatic potential.

**Figure supplement 1—source data 1.** Data for the graphs in the figure.

**Figure supplement 2.** Immunophenotypic analysis of primary tumors and liver metastases in the orthotopic CT26^LM3^ model.

**Figure supplement 2—source data 1.** Data for the graphs in the figure.

MRI analysis at day 21 after CRC challenge revealed that, in comparison with liver metastases observed in NaCl-treated controls, the lesions produced by MC38- or MC38^Ifnar1_KO^ cells in IFNα-treated mice were similarly reduced in number and size (*Figure 4A and C–D*) and this resulted in comparable mouse survival rates (*Figure 4E*) in the absence of apparent myelotoxicity (*Figure 4—figure supplement 2A*). These data support the hypothesis that in our experimental setting the continuous IFNα administration has no direct antiproliferative activity towards CRC cells consistent with our previous reported data (*Figure 4—figure supplement 1A-C*; *Catarinella et al., 2016*).

Next, we crossed *Ifnar1*-floxed mice (termed *Ifnar1^fl/fl^*) (*Prigge et al., 2015*) with transgenic mice selectively expressing Cre recombinase in parenchymal and non-parenchymal liver cells (*Gerl et al., 2015*; *Postic et al., 1999*; *Wang et al., 2010*; *Figure 4—figure supplement 1D*). Cell type-specific recombination was confirmed by crossing each parental mouse line with Rosa26-ZsGreen reporter mice. Note that by crossing the parental Cre-expressing lines with Rosa26-ZsGreen reporter mice (*Madisen et al., 2010*) the resultant mice showed specific recombination in most hepatocytes (identified by morphology), and liver fibroblast (GFP⁺/PDGFRβ⁺), with about 98.2 ± 0.72% hepatic stellate cells that co-expressed GFP⁺ and PDGFRβ⁺ signals (*Figure 4—figure supplement 1E, F*). Similarly, hepatic DCs (GFP⁺/CD11c⁺) had 94.17 ± 2.16% colocalization with GFP, while the colocalization percentage of F4/80⁺ KCs or LCMs (GFP⁺/F4/80⁺) was 78.14 ± 5.03% (*Figure 4—figure supplement 1E,F*; *Blériot and Ginhoux, 2019*; *Karmaus and Chi, 2014*; *Madisen et al., 2010*). Finally, HECs, including LSECs, (GFP⁺/CD31⁺) showed 85.3 ± 5.03% colocalization (*Figure 4—figure supplement 1E, F*), with no expression of GFP signals in cells other than CD31⁺. *Ifnar1^fl/fl^* control mice and mice lacking *Ifnar1* in hepatocytes (termed *Alb^Ifnar1_KO^*), hepatic stellate cells (termed *Pdgfrb^Ifnar1_KO^*), *Itgax⁺* (CD11c) DCs/KCs/LCMs (termed *Itgax^Ifnar1_KO^*) or *Cdh5⁺* + cells (termed *VeCad^Ifnar1_KO^*) were intramesenterically injected with MC38 cells 7 days after NaCl or IFNα therapy initiation and did not show signs of hematotoxicity during IFNα infusion (*Figure 4—figure supplement 2B*). Metastatic growth was assessed by MRI at day 21 (*Figure 4B*). Loss of *Ifnar1* on hepatocytes, hepatic stellate cells or DCs/KCs/LCMs did not significantly alter the anti-metastatic activity of IFNα treatment (*Figure 4C, D and F* and *Figure 4—figure supplement 2C*). By contrast, the depletion of *Ifnar1* on HECs allowed the lesions to grow undisturbed (*Figure 4B*). Indeed, *VeCad^Ifnar1_KO^* mice treated with either NaCl or IFNα displayed very similar numbers and sizes of hepatic lesions (*Figure 4C and D*) or survival rates (*Figure 4F*), indicating that the antimetastatic properties of IFNα requires *Ifnar1* signaling on HECs. *VeCad^Ifnar1_KO^* mice exhibited increased tumor burden (*Figure 4D*) and mortality rates (*Figure 4—figure supplement 2D*) when compared to NaCl-treated *Ifnar1^fl/fl^* mice, suggesting that hepatic endothelial *Ifnar1* signaling exerts significant anti-tumor activity even in the context of physiologic endogenous intrahepatic levels of type I interferons. Furthermore, histological analysis of hepatic CRC lesions from

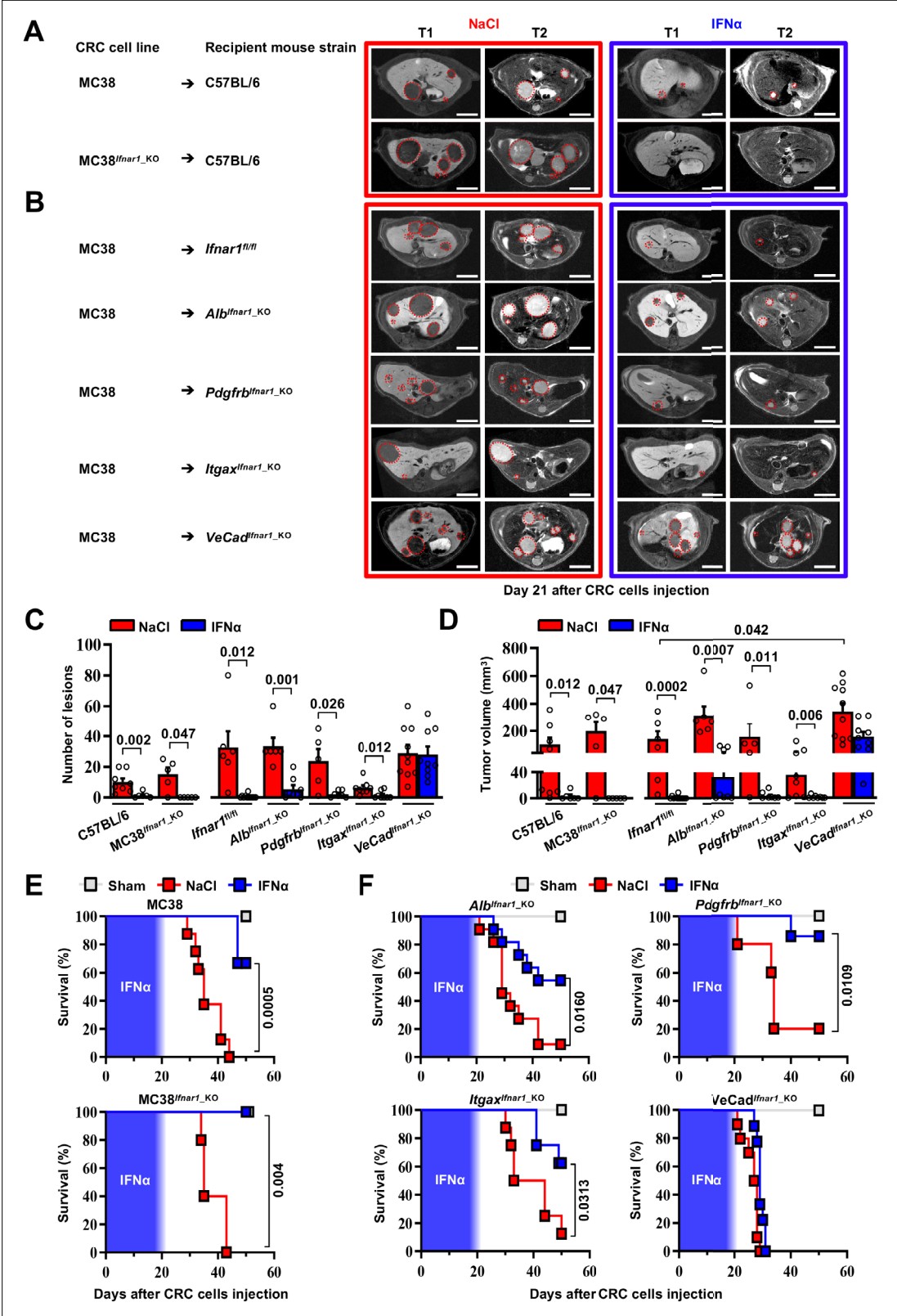

**Figure 4.** HECs mediate the antimetastatic activity of IFNα. (**A**) Representative hepatic contrast-enhanced MRI of wild-type mice (C57BL/6) at day 21 after injection of 5x10⁴ MC38 cells or 5x10⁴ MC38^{Ifnar1_KO} cells. Treatment with NaCl (red frame) and IFNα (blue frame) was initiated 7 days prior to intrasplenic injection of CRC cells. Red dashed lines highlight CRC liver metastases, characterized as hypointense and slightly-hyperintense regions in T1- and T2-weighted sequences, respectively. Scale bar = 5 mm. The number of mice for each experimental condition is also indicated.

*Figure 4 continued on next page*

**eLife** Research article

Cancer Biology | Immunology and Inflammation

*Figure 4 continued*

(**B**) Representative hepatic contrast-enhanced MRI at day 21 after MC38 intrasplenic injection of *Ifnar1*$^{fl/fl}$ mice and mice lacking *Ifnar1* on hepatocytes (*Alb*$^{Ifnar1\_KO}$), hepatic stellate cells (*Pdgfrb*$^{Ifnar1\_KO}$), CD11c-expressing DCs/KCs/LMCs (*Itgax*$^{Ifnar1\_KO}$) and endothelial cells (*VeCad*$^{Ifnar1\_KO}$) injected with 5x10$^4$ MC38 cells. Treatment with NaCl (red frame) and IFNα (blue frame) was initiated 7 days before MC38 cells intrasplenic injection. Red dashed lines highlight CRC liver metastases, characterized as hypointense and slightly-hyperintense regions in T1- and T2-weighted sequences, respectively. Scale bar = 5 mm. Number of mice for each experimental condition are also indicated. (**C**) Quantification of the number of hepatic lesions by MRI analysis, 21 days after CRC cells injection, of NaCl and IFNα-treated mice found in C57BL/6 mice, *Ifnar1*$^{fl/fl}$ mice and all conditional *Ifnar1*_KO mouse models described in (**A**) and (**B**). Mean values ± SEM are shown; p-values were calculated by Mann-Whitney test. (**D**) Quantification of total tumor volume of liver metastases by MRI analysis at day 21 after CRC intrasplenic injection of NaCl- and IFNα-treated mice found in C57BL/6 mice, *Ifnar1*$^{fl/fl}$ mice and all conditional *Ifnar1*_KO mouse models analyzed. Mean values ± SEM are shown; p-values were calculated by Mann-Whitney test. (**E**) Kaplan-Meier survival curves of C57BL/6 mice injected with MC38 cells (top) or MC38$^{Ifnar1\_KO}$ cells (bottom) described in (**A**). Sham injected animals (n=3) were used as control. The blue pattern indicates the time frame of IFNα ip release; p-values were calculated by log-rank/Mantel-Cox test. (**F**) Kaplan-Meier survival curves of the indicated groups of mice described in (**B**). Sham injected animals (n=3 per group) were used as control. The blue pattern indicates the time frame of IFNα ip release. The total number of mice for each experimental group were: NaCl-C57BL/6 n=8; IFNα-C57BL/6 n=6; NaCl-MC38$^{Ifnar1\_KO}$-C57BL/6 n=5; IFNα-MC38$^{Ifnar1\_KO}$-C57BL/6 n=6; NaCl-*Ifnar1*$^{fl/fl}$ n=6; IFNα-*Ifnar1*$^{fl/fl}$ n=11; NaCl-*Alb*$^{Ifnar1\_KO}$ n=6; IFNα-*Alb*$^{Ifnar1\_KO}$ n=8; NaCl-*Pdgfrb*$^{Ifnar1\_KO}$ n=5; IFNα-*Pdgfrb*$^{Ifnar1\_KO}$ n=7; NaCl-*Itgax*$^{Ifnar1\_KO}$ n=8; IFNα-*Itgax*$^{Ifnar1\_KO}$ n=8 and NaCl-*VeCad*$^{Ifnar1\_KO}$ n=10; IFNα-*VeCad*$^{Ifnar1\_KO}$ n=9. Data pooled from at least two independent experiments of each experimental group; p-values were calculated by log-rank/Mantel-Cox test.

The online version of this article includes the following source data and figure supplement(s) for figure 4:

**Source data 1.** Data for the graphs in the figure.

**Figure supplement 1.** Characterization of conditional *Ifnar1* knockout MC38 cells and liver specific *Ifnar1* knockout mouse models.

**Figure supplement 1—source data 1.** Original gel file and uncropped gel with labeling of panel (**A**).

**Figure supplement 1—source data 2.** Data for the graphs in the figure.

**Figure supplement 1—source data 3.** Original gel file and uncropped gel with labeling of panel (**D**).

**Figure supplement 2.** LSECs mediate the antimetastatic activity of IFNα.

**Figure supplement 2—source data 1.** Data for the graphs in the figure.

**Figure supplement 3.** Characterization of liver metastases developed in *VeCad*$^{Ifnar1\_KO}$ mice at day 21 after intramesenteric MC38 cells injection.

**Figure supplement 3—source data 1.** Data for the graphs in the figure.

NaCl- and IFNα-treated *VeCad*$^{Ifnar1\_KO}$ mice euthanized at day 21 after MC38 intramesenteric injection indicated that these tumors resembled NaCl-treated *Ifnar1*$^{fl/fl}$ lesions, showing signs of angiogenesis (as denoted by CD34 positivity) and similar content of F4/80$^+$ macrophages and CD3$^+$ T cells within the intrahepatic CRC foci (*Figure 4—figure supplement 3A, B*).

## Continuous IFNα administration limits trans-sinusoidal migration of CRC cells by strengthening the liver vascular barrier

We next took advantage of fluorescence-based techniques to investigate the initial steps of liver colonization. First, we assessed the intrahepatic localization of GFP-expressing MC38 cells (MC38$^{GFP}$) (*Talamini et al., 2021*) that were intramesenterically challenged 5 min earlier. Most MC38$^{GFP}$ cells in *Ifnar1*$^{fl/fl}$ or *VeCad*$^{Ifnar1\_KO}$ mice appeared physically trapped at the beginning of the sinusoidal circulation in both mouse lineages. This was evidenced by the close contact of MC38$^{GFP}$ cells with LYVE-1-expressing LSECs in the proximity of the portal tracts (*Figure 5—figure supplement 1A*). Further, MC38$^{GFP}$ cells arrested where the sinusoidal diameter (*Figure 5—figure supplement 1B*) is smaller than their own (12±0.1 μm), similar to what we previously reported (*Catarinella et al., 2016*). As the process of trans-sinusoidal migration – a critical limiting step in the metastatic cascade – is known to occur within 24 hr of CRC challenge (*Chambers et al., 2002*; *Valastyan and Weinberg, 2011*; *Wolf et al., 2012*), the intrahepatic number and localization of MC38$^{GFP}$ cells were then studied at this time point. Confocal IF quantification revealed that, compared to NaCl-treated *Ifnar1*$^{fl/fl}$ animals, MC38$^{GFP}$ cells were about ~twofold less abundant in IFNα-treated *Ifnar1*$^{fl/fl}$ mice and ~threefold more abundant in *VeCad*$^{Ifnar1\_KO}$ mice treated with NaCl or IFNα (*Figure 5A* top). Moreover, confocal 3D reconstructions of liver sinusoids from IFNα-treated *Ifnar1*$^{fl/fl}$ mice unveiled that by 24 hr most MC38$^{GFP}$ cells localize intravascularly (i.e. they did not invade the liver parenchyma), while in NaCl-treated *Ifnar1*$^{fl/fl}$ controls and in NaCl- or IFNα-treated *VeCad*$^{Ifnar1\_KO}$ mice only few MC38$^{GFP}$ cells remain within the liver vasculature (i.e. they invaded the liver parenchyma) (*Figure 5A* bottom, 5B and *Figure 5—videos*

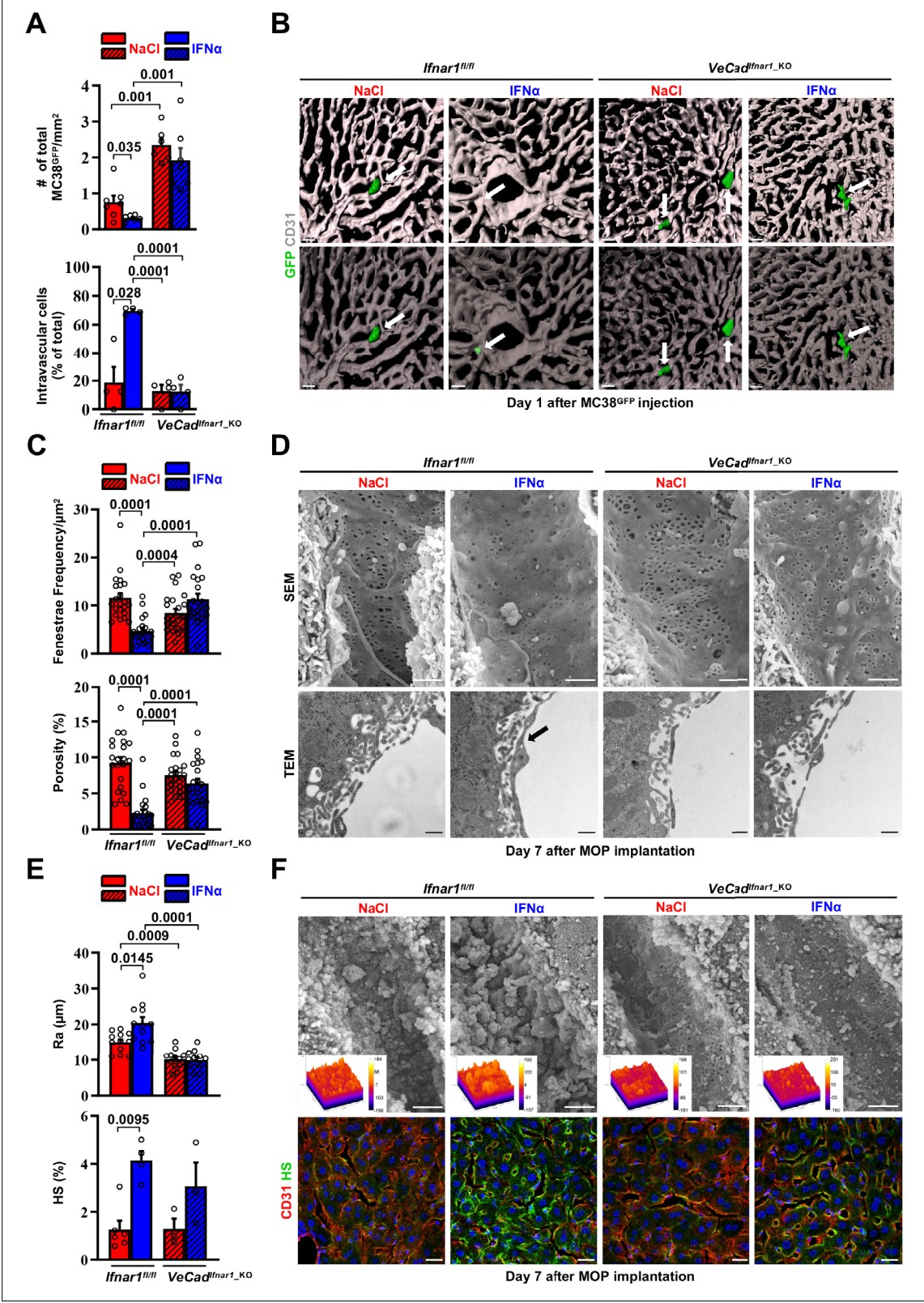

**Figure 5.** Continuous IFNα administration limits trans-sinusoidal migration of CRC cells by strengthening the hepatic vascular barrier. (**A**) Total number of MC38GFP cells per area (top) and total number of intravascular MC38GFP cells per tissue area (bottom). The total hepatic area was approximately 5 mm² for each experimental group. Intravascular localization was measured on 20 x images, 5 randomly selected images per mouse (n=3 per group). Mean values ± SEM are shown; p-values were calculated by Mann-Whitney test. (**B**) Confocal reconstruction of the liver vasculature from *Ifnar1*fl/

*Figure 5 continued on next page*

*Figure 5 continued*

*fl* and *VeCad*[Ifnar1_KO] mice 24 hr after MC38[GFP] cells (green) intramesenteric vein injection in mice that were treated with NaCl or IFNα for 7 days. CD31 is shown in grey. To allow visualization of intravascular MC38[GFP] cells and to enhance image clarity, the transparency of the sinusoidal rendering was increased up to 80% (bottom). Scale bars = 20μm. (**C**) Fenestrae frequency histogram (top) and percentage of vessel porosity (the percentage of liver endothelial surface area occupied by fenestrae, bottom) in liver sections of control *Ifnar1*[fl/fl] and *VeCad*[Ifnar1-KO] mice treated for 7 days with continuous NaCl or IFNα therapy. Quantification was performed on 17.000 x SEM images, 10 randomly selected images per mouse (n=3 per group). A total area of approximately 720 μm² of sinusoidal surface was analyzed for each mouse. Mean values ± SEM are shown; p-values were calculated by Mann-Whitney test. (**D**) Representative scanning electron micrographs (SEM, top) and transmission electron micrographs (TEM, bottom) images from liver sections of mice described in (**C**), showing hepatic fenestrations and endothelial features. Arrow indicates the increased endothelial thickness observed after continuous IFNα therapy in *Ifnar1*[fl/fl] mice. SEM scale bars = 1 μm; TEM scale bars = 500 nm. (**E**) Quantification of the arithmetical mean deviation or Ra coefficient (top) and the percentage of hepatic area positive for HS staining (bottom) on *Ifnar1*[fl/fl] and *VeCad*[Ifnar1_KO] livers treated with continuous NaCl or IFNα therapy for 7 days. Quantification was performed on at least 3 *Ifnar1*[fl/fl] and *VeCad*[Ifnar1_KO] livers per group. Mean values ± SEM are shown; p-values were calculated by Mann-Whitney test. (**F**) Liver sinusoidal endothelial glycocalyx (GCX) visualization by scanning electron micrographs (SEM, top) from *Ifnar1*[fl/fl] and *VeCad*[Ifnar1_KO] livers that were perfused with lanthanum nitrate (a heavy metal that allows GCX visualization by stabilizing negatively charged GCX structures) 7 days after continuous NaCl or IFNα therapy. Scale bars = 1 μm. Inserts display a representation of the 3D topographic surface of a selected area within the liver sinusoid of each experimental condition. Representative immunofluorescence micrographs of Heparan sulfate (HS; green) and CD31 (red) staining from *Ifnar1*[fl/fl] and *VeCad*[Ifnar1_KO] mice treated with continuous NaCl or IFNα therapy for 7 days, showing increased HS accumulation after IFNα therapy only in *Ifnar1*[fl/fl] mice (bottom). Hoechst (blue) was used for nuclear counterstaining. Scale bars = 20μm.

The online version of this article includes the following video, source data, and figure supplement(s) for figure 5:

**Source data 1.** Data for the graphs in the figure.

**Source data 2.** High-magnification immunofluorescence images of each channel.

**Figure supplement 1.** Continuous IFNα administration induces hepatic endothelial capillarization strengthening the liver vascular barrier.

**Figure supplement 1—source data 1.** Data for the graphs in the figure.

**Figure supplement 1—source data 2.** Collagen type IV.

**Figure supplement 1—source data 3.** Laminin.

**Figure supplement 1—source data 4.** ICAM1.

**Figure supplement 1—source data 5.** E-Selectin.

**Figure supplement 2.** Continuous IFNα administration induces hepatic endothelial capillarization that is reversed after discontinuation of IFNα therapy.

**Figure supplement 2—source data 1.** LYVE-1.

**Figure supplement 2—source data 2.** Data for the graphs in the figure.

**Figure 5—video 1.** 3D reconstruction of a representative extravasated MC38[GFP] cell found within the liver parenchyma of a NaCl-treated *Ifnar1*[fl/fl] mouse.

https://elifesciences.org/articles/80690/figures#fig5video1

**Figure 5—video 2.** 3D reconstruction of an intravascular MC38[GFP] cell found within a blood vessel of an IFNα-treated *Ifnar1*[fl/fl] mouse.

https://elifesciences.org/articles/80690/figures#fig5video2

**Figure 5—video 3.** 3D reconstruction of an extravasated MC38[GFP] cell found within the liver parenchyma of a NaCl-treated *VeCad*[Ifnar1_KO] mouse.

https://elifesciences.org/articles/80690/figures#fig5video3

**Figure 5—video 4.** 3D reconstruction of extravasated MC38[GFP] cells found within the liver parenchyma of an IFNα-treated *VeCad*[Ifnar1_KO] mouse.

https://elifesciences.org/articles/80690/figures#fig5video4

*1–4*). These results indicate that HECs, including LSECs, negatively control trans-sinusoidal CRC migration upon IFNα sensing.

To unravel phenotypic modifications associated with such antitumor function of HECs, including LSECs, the liver microvasculature of NaCl- or IFNα-treated *Ifnar1*[fl/fl] and *VeCad*[Ifnar1_KO] mice was analyzed by scanning electron microscopy (SEM) and transmission electron microscopy (TEM). IFNα treatment of *Ifnar1*[fl/fl] mice significantly decreased the frequency of sinusoidal fenestrae and the overall porosity of LSECs (*Figure 5C and D* top), while it increased: (i) the endothelial thickness (*Figure 5—figure supplement 1D*), (ii) the space of Disse density (an indirect measure of hepatocyte microvilli density) (*Figure 5—figure supplement 1D*; *Gissen and Arias, 2015*), (iii) the subendothelial deposition of collagen fibrils (*Figure 5—figure supplement 1D*) and (iv) the appearance of a basal lamina (*Figure 5D* bottom and *Figure 5—figure supplement 1D*). These results were corroborated by immunofluorescence analysis assessing an enhanced perivascular expression of Collagen type IV and Laminin (*Figure 5—figure supplement 1E, F*), two components of the basal lamina previously shown to form a barrier against tumor cell invasion (*Mak and Mei, 2017*; *Tanjore and Kalluri, 2006*).

By contrast, IFNα treatment of *VeCad*[Ifnar1_KO] mice failed to significantly support these changes, leaving the liver microvasculature of these animals highly similar to that of liver metastases-permissive NaCl-treated *Ifnar1*[fl/fl] controls (*Figure 5C and D* and *Figure 5—figure supplement 1D-F*). Moreover, IFNα treatment of *Ifnar1*[fl/fl] mice significantly increased the expression of LYVE-1, a marker of hepatic capillarization (*Pandey et al., 2020*; *Wohlfeil et al., 2019*). By contrast, IFNα treatment of *VeCad*[Ifnar1_KO] mice showed no effect (*Figure 5—figure supplement 2A,B*).

Next, we evaluated the status of the vascular glycocalyx (GCX), a fibrous network of glycoproteins and proteoglycans that lines the LSECs and projects intraluminally (*Reitsma et al., 2007*). Notably, enhanced GCX deposits can act as a repulsive barrier that prevents tumor cell interactions with endothelial cells, adhesion molecules or chemokines have been previously identified as negative correlates of transendothelial migration (*Glinskii et al., 2005*; *Mitchell and King, 2014*; *Offeddu et al., 2021*; *Wilkinson et al., 2020*). Continuous IFNα treatment modified this network as well, increasing its thickness (*Figure 5E and F* top) and the expression of one of its major components, the heparan sulfate (HS) (*Reitsma et al., 2007*; *Figure 5E and F* bottom). Of note, *VeCad*[Ifnar1_KO] mice displayed reduced GCX thickness independently of NaCl- or IFNα-treatment (*Figure 5E and F*). Additionally, we evaluated the vascular and perivascular status of cell adhesion molecules such as selectins and integrins, which have been positively associated with the transendothelial migration of tumor cells (*Glinskii et al., 2005*; *Wilkinson et al., 2020*). The expression of ICAM1, E-Selectin (CD62E) (*Figure 5—figure supplement 1G,H*) and the integrins ITGB2 (CD18) or ITGA4 (CD49d) (*Figure 5—figure supplement 1C*) was up-regulated in IFNα-treated *Ifnar1*[fl/fl] controls, while significantly reduced or attenuated in IFNα-treated *VeCad*[Ifnar1_KO] mice. The notion that a more modest upregulation of some these markers was still evident in the latter mice may reflect the capacity of liver cells other than HECs to respond to IFNα. Altogether, these results indicate that numerous phenotypic modifications of the liver microvasculature previously associated with the deficient extravasation of both normal and transformed cells of different origin (*Guidotti et al., 2015*; *Valastyan and Weinberg, 2011*) also occur because of continuous IFNα sensing by HECs, including LSECs. Notably, these microvascular modifications were reverted after the discontinuation of IFNα therapy with no impact on long-term liver functionality/viability (*Figure 5—figure supplement 2C-I*).

## HECs acquire an antimetastatic transcriptional profile upon continuous IFNα sensing

To confirm the above-mentioned data and to shed new light on the transcriptional changes that HECs adopt to limit CRC trans-sinusoidal migration, we performed RNA-seq analyses on CD31⁺ endothelial cells isolated from the liver of *Ifnar1*[fl/fl] or *VeCad*[Ifnar1_KO] mice 7 days after NaCl or IFNα treatment (*Figure 6—figure supplement 1A*). Using SEM to assess the % of CD31⁺ cells bearing the typical sinusoidal fenestrae, we determined that our preparations contain ~96% of bona-fide LSECs (*Figure 6—figure supplement 1B*), consistently to previous reports (*Liu et al., 2011*; *Su et al., 2021*). When compared to HECs isolated from NaCl-treated *Ifnar1*[fl/fl] mice, HECs derived from IFNα-treated animals of the same lineage showed 381 transcripts that were differentially expressed (*Figure 6A*). As expected, many of these up-regulated transcripts belonged to the ISG family, including *Irf7, Irf9, Mx1, Mx2, Isg15, Stat1,* and *Oasl1* (*Figure 6A and B*). Pre-ranked gene set enrichment analyses (GSEA) of IFNα-treated LSECs also revealed a significant enrichment of transcripts involved in interferon signaling or in the induction of varying cytokines and chemokines (*Figure 6C*). Several transcripts related to the ECM/GCX organization or the cell-cell/cell-matrix adhesion pathways were upregulated as well (*Figure 6B and D*). Of note, the expression of *Itga4* and *Itgb2* – previously shown to be increased by IFNα treatment at the protein level (*Figure 5—figure supplement 1C*) – was also enhanced at the transcriptional level (*Figure 6B*). A similar association did not hold true for *Icam1* and *Sele*, suggesting that the increased protein expression we observed earlier (*Figure 5—figure supplement 1G*) occurred independently of transcriptional activity (*Figure 6B*). Notably, GSEA also identified gene sets involved in the IFNα-dependent activation of innate and adaptive immune responses or in TCR-dependent signaling pathways (*Figure 6—figure supplement 1C*).

Keeping the HECs transcriptional profile of NaCl-treated *Ifnar1*[fl/fl] mice as a point of reference, a total of 566 genes were differentially expressed (DEGs) in HECs isolated from NaCl-treated *VeCad*[Ifnar1_KO] mice, of which 373 (mostly ISGs and genes involved in the immune response or in the antigen processing) were downregulated (*Figure 6A–C*). These latter results indirectly suggest that – when

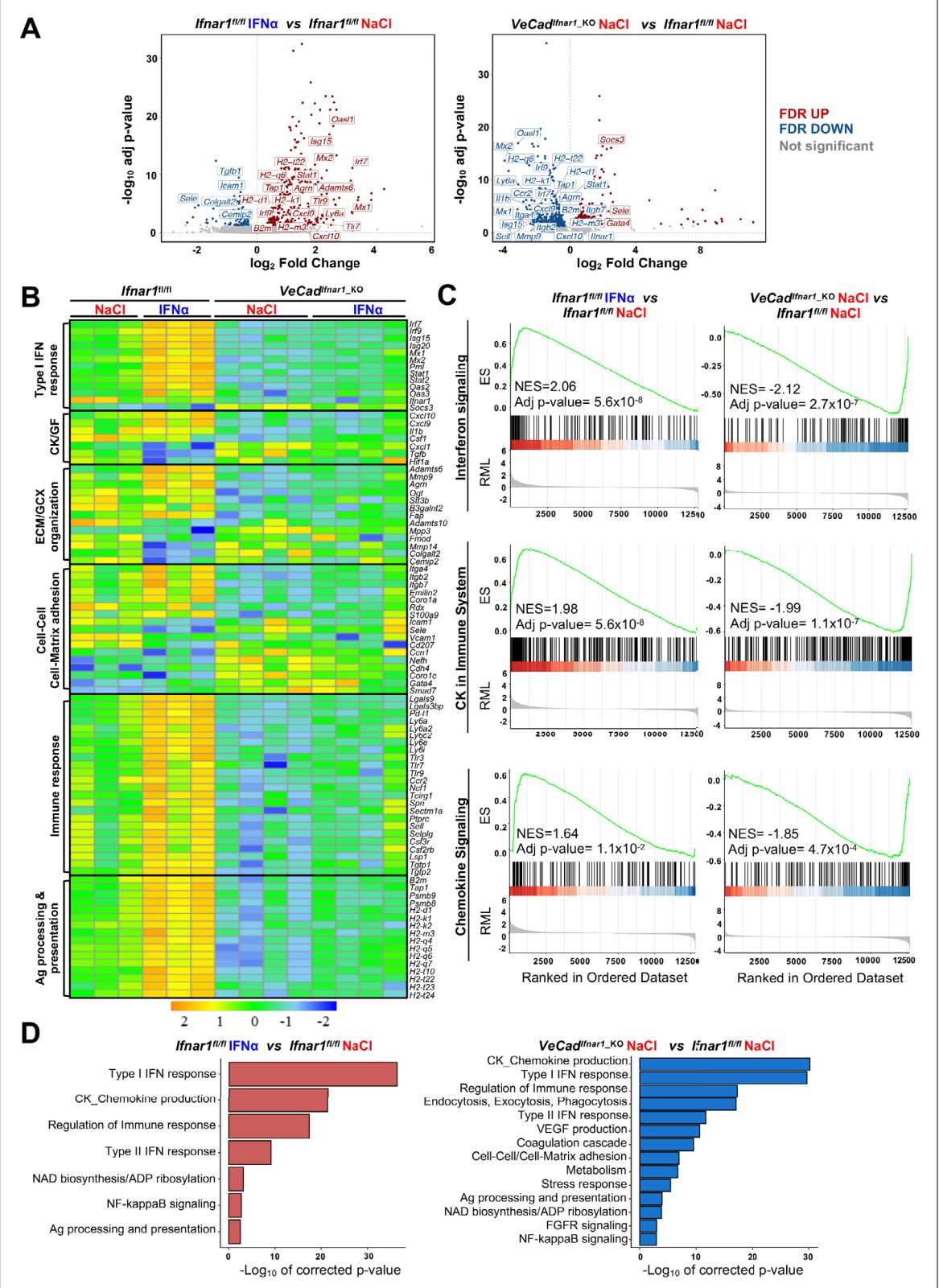

**Figure 6.** HECs acquire an antimetastatic transcriptional profile upon continuous IFNα sensing. (**A**) Volcano plots of differential gene expression (DGE) results obtained from the comparisons between HECs-derived from *Ifnar1^(fl/fl)* mice treated with NaCl and IFNα for 7 days (left) and between HECs-derived from *VeCad^(Ifnar1_KO)* and *Ifnar1^(fl/fl)* NaCl-treated mice (right) (*Ifnar1^(fl/fl)* n=3; *VeCad^(Ifnar1_KO)* n=4). Significant false discovery rate (FDR <0.05) up- and down-regulated genes are highlighted in red and blue colors, respectively, not significant genes are depicted in grey. (**B**) Heatmap of the expression

*Figure 6 continued on next page*

*Figure 6 continued*

values (log2-transformed rpkm) of manually selected genes retrieved from differentially regulated pathways. (**C**) Pre-ranked Gene Set Enrichment Analysis (GSEA) enrichment plots of the indicated pathways between CD31$^+$ cells from *Ifnar1*$^{fl/fl}$-IFNα and *Ifnar1*$^{fl/fl}$-NaCl (left) and between *VeCad*$^{Ifnar1\_KO}$-NaCl and *Ifnar1*$^{fl/fl}$-NaCl-treated animals (right). (**D**) Bar charts showing the adjusted p-values (-log10 transformed) of selected pathways from the enrichment analysis performed on comparisons between *Ifnar1*$^{fl/fl}$-IFNα and *Ifnar1*$^{fl/fl}$-NaCl CD31$^+$ cells (left graph) and between *VeCad*$^{Ifnar1\_KO}$-NaCl and *Ifnar1*$^{fl/fl}$-NaCl CD31$^+$ cells (right graph).

The online version of this article includes the following source data and figure supplement(s) for figure 6:

**Source data 1.** Gene Ontology (GO) of biological process enrichment analysis of all differentially expressed genes.

**Figure supplement 1.** HECs acquire an antimetastatic transcriptional profile upon continuous IFNα sensing.

**Figure supplement 1—source data 1.** Data for the graphs in the figure.

compared to HECs capable of sensing low levels of endogenous type I IFNs, as those present in NaCl-treated *Ifnar1*$^{fl/fl}$ mice – LSECs devoted of *Ifnar1* may be less prepared to stimulate innate and adaptive immunity (*Figure 6B*). The downregulation of transcripts involved in cell-cell adhesion molecules and matrix remodeling (*Figure 6—figure supplement 1D*) further suggests a relative unpreparedness of *VeCad*$^{Ifnar1\_KO}$ HECs at accommodating changes that may confer protection against CRC trans-sinusoidal cell migration. Along these lines, the upregulation of the transcripts for *Gata4* -a master-regulator of liver sinusoidal differentiation which leads to liver fibrosis deposition upon its loss (*Winkler et al., 2021*)- and *Smad7* – an inhibitor of transforming growth factor-beta (TGF-β)-dependent subendothelial matrix deposition causative of sinusoidal capillarization (*Tauriello et al., 2018*) – in HECs isolated from NaCl-treated *VeCad*$^{Ifnar1\_KO}$ mice (*Figure 6A*) could be interpreted as a diminished capacity to shape a vascular antimetastatic barrier. Finally, Gene Ontology (GO) analysis of HECs confirmed that *Ifnar1*-proficient, but not *Ifnar1*-deficient, HECs upregulate transcriptional pathways involved in the production of immunostimulatory cytokines and chemokines, the capacity to process and present antigens or the regulation of immune responses (*Figure 6D*). Altogether, the data support the hypothesis that, upon IFNα sensing, HECs and particularly LSECs not only acquire a transcriptional profile that can reinforce their barrier function, but they may also enhance HECs/LSECs immunostimulatory functions contributing to antitumor activity.

## Continuous IFNα sensing improves immunostimulatory properties of HECs to provide long-term tumor protection

First, HECs/LSECs isolated from the liver of *Ifnar1*$^{fl/fl}$ or *VeCad*$^{Ifnar1\_KO}$ mice 7 days after continuous NaCl or IFNα treatment were assessed for the relative surface protein expression of MHC-I, CD86 (a costimulatory molecule *Katz et al., 2004*) or the interleukin 6 receptor alpha (IL-6RA, a molecule that LSECs use to properly cross-prime antigens to naïve CD8$^+$ T cells *Böttcher et al., 2014*). Following IFNα treatment, *Ifnar1*-bearing LSECs significantly increased MHC-I, CD86 and IL-6RA expression (*Figure 7—figure supplement 1A*), while no induction was detected in *Ifnar1*-negative LSECs (*Figure 7—figure supplement 1A*). We then analyzed the ability of IFNα-treated LSECs or splenic DCs (sDCs) from *Ifnar1*$^{fl/fl}$ and *VeCad*$^{Ifnar1\_KO}$ mice to stimulate the cross-priming of naive CD8$^+$ T cells in vitro. To this end, viable CD31$^+$ HECs and CD11c$^+$ sDCs were isolated and purified (*Figure 7—figure supplement 2A*, B). sDCs were cultured to acquire mature CD8α$^+$ (~25%) or plasmacytoid (45%–50%) phenotypes endowed with cross-priming capacity (*Figure 7—figure supplement 2C, D*; *Fu et al., 2020*). HECs and sDCs from *Ifnar1*$^{fl/fl}$ or *VeCad*$^{Ifnar1\_KO}$ mice previously pulsed with the SIINFEKL peptide or soluble ovalbumin (sOVA) in the presence or absence of either IFNα or NaCl were then co-cultured with naive OT-I CD8$^+$ T cells and their relative cross-priming capacity was defined by the percentage of these latter cells to express both CD44 and IFNγ (*Figure 7—figure supplement 1B*). IFNα stimulation of *Ifnar1*-bearing HECs (HECs from *Ifnar1*$^{fl/fl}$ mice or sDCs from *Ifnar1*$^{fl/fl}$ and *VeCad*$^{Ifnar1\_KO}$ mice) pulsed with SIINFEKL or sOVA promptly increased their cross-priming capacities, while the same IFNα treatment failed to do so in *Ifnar1*-negative cells (HECs from *VeCad*$^{Ifnar1\_KO}$ mice) (*Figure 7A and B* and *Figure 7—figure supplement 1B, C*). Once exposed to IFNα and pulsed with sOVA, HECs and sDCs from *Ifnar1*$^{fl/fl}$ mice cross-primed naïve OT-I CD8$^+$T cells to a similar extent (*Figure 7B* and *Figure 7—figure supplement 1C*), highlighting once more the immunostimulating potential of IFNα treatment on HECs, including LSECs. We also evaluated the splenic composition of central memory T cell populations (Tcm, CD8$^+$CD44$^+$CD62L$^+$) (*Figure 7C* and *Figure 7—figure*

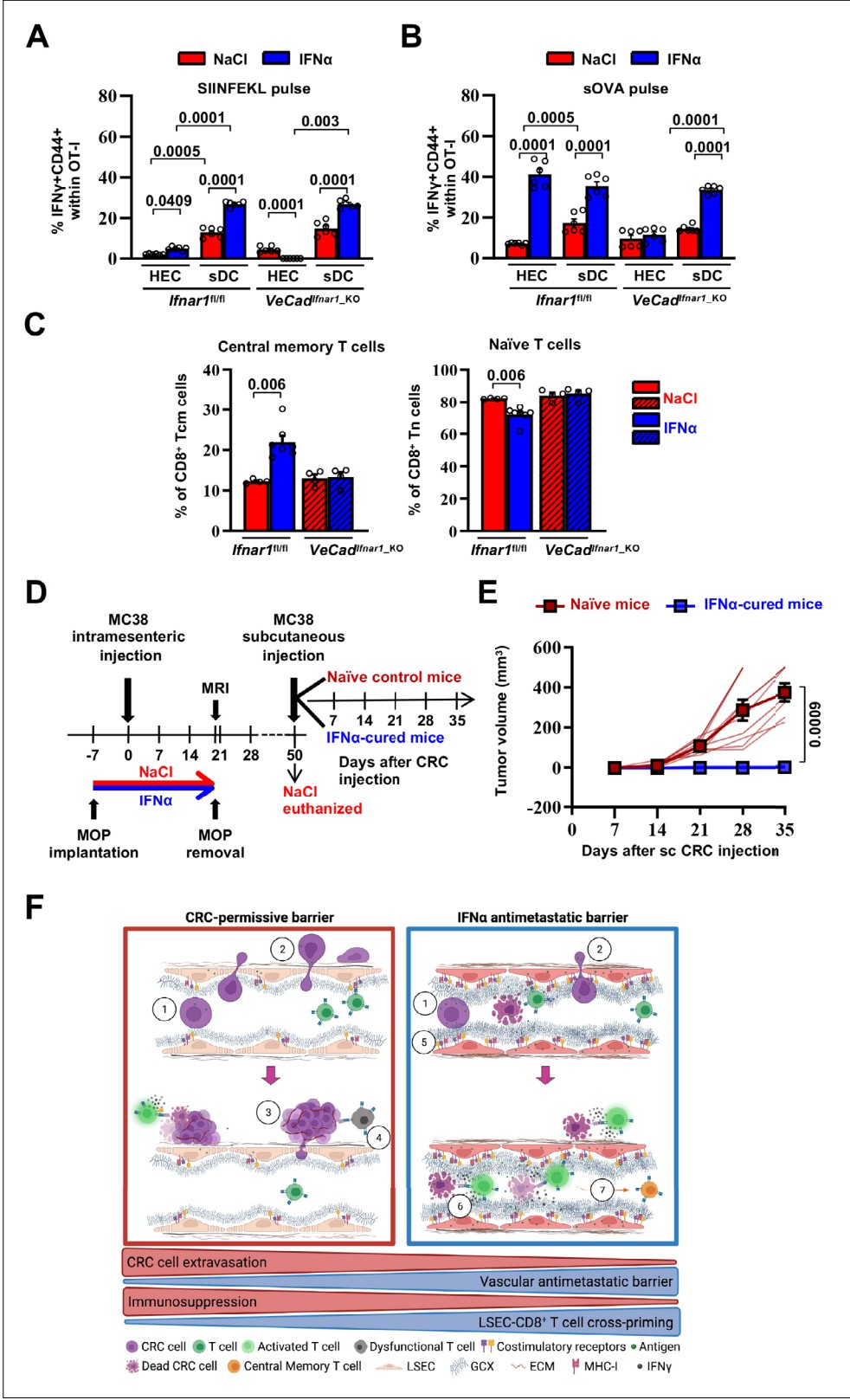

**Figure 7.** Continuous IFNα sensing improves immunostimulatory properties of HECs to provide long-term tumor protection. (**A–B**) Quantification of the percentage of OT-I CD8+ T cells expressing CD44+IFNγ+ generated after the co-cultured with HECs, including LSECs, or sDCs isolated from *Ifnar1*fl/fl and *VeCad*Ifnar1-KO mice and stimulated with SIINFEKL peptide (**A**) or soluble OVA protein (sOVA) (**B**) in the presence of NaCl or IFNα. Mean values ± SEM are

*Figure 7 continued on next page*

*Figure 7 continued*

shown; p-values were calculated by two-way ANOVA test from two independent experiments with three biological replicates each. (**C**) Quantification of the percentage of splenic T central memory (Tcm, CD8$^+$CD44$^+$CD62L$^+$) and naïve T cells (Tn, CD8$^+$CD44$^-$CD62L$^+$) populations 21 days after intramesenteric MC38 cells injection into NaCl- or IFNα-treated *Ifnar1*$^{fl/fl}$ and *VeCad*$^{Ifnar1\_KO}$ mice. Quantification was performed on least 4 mice per group. Mean values ± SEM are shown; p-values were calculated by Mann-Whitney test. (**D**) Schematic representation of the experimental procedure used for tumor rechallenge. IFNα-*Ifnar1*$^{fl/fl}$-cured mice and naïve *Ifnar1*$^{fl/fl}$ mice were subcutaneously rechallenged with 5x10$^3$ MC38 cells and received no further treatment. Tumor growth was monitored weekly for 35 days. Note that we used aged-matched *Ifnar1*$^{fl/fl}$ naïve mice because NaCl-treated mice had to be euthanized for ethical reasons by day 50. (**E**) Kinetics of subcutaneous tumor growth in naïve (n=10) and IFNα-cured (n=7) mice. Both, mean tumor volume and individual animal measurements are shown. Mean values ± SEM are shown; p-values were calculated by two-way ANOVA test. (**F**) Schematic model: CRC cells emerging from the primary tumor reach the hepatic sinusoids via the portal circulation and arrest - mostly because of size constrains - at the portal side of the sinusoidal circulation (1), CRC cells trans-sinusoidally migrate into the liver parenchyma (2) and develop micro-metastases that will eventually grow overtime (3; red frame), promoting the generation of an immunosuppressive microenvironment leading to dysfunctional T cells (4; red frame). Conversely, IFNα therapy (blue frame), by modifying LSECs porosity, thickness, deposition of basal membrane and GCX depth, builds up a vascular antimetastatic barrier (5), that impairs CRC trans-sinusoidal migration, promoting intravascular containment of invading tumor cells (6) that together with IFNα-mediated increased cross-presentation and cross-priming by HECs/LSEC, will lead to naïve CD8$^+$ T cell activation and secondary generation of long-term antitumor immunity and protection from secondary tumor challenge (7). Created with https://biorender.com/.

The online version of this article includes the following source data and figure supplement(s) for figure 7:

**Source data 1.** Data for the graphs in the figure.

**Figure supplement 1.** Continuous IFNα sensing improves immunostimulatory properties of HECs to provide long-term tumor protection.

**Figure supplement 1—source data 1.** Data for the graphs in the figure.

**Figure supplement 2.** Isolation and flow cytometry characterization of HECs and sDCs.

---

*supplement 1D*) as a proxy of potential systemic memory responses against tumor antigens (*Sallusto et al., 2004*; *Stone et al., 2009*; *Yu et al., 2019*). Splenic naive T cells (Tn, CD8$^+$CD44$^-$CD62L$^+$) were also evaluated. Looking at *Ifnar1*$^{fl/fl}$ or *VeCad*$^{Ifnar1\_KO}$ mice continuously treated with NaCl or IFNα and euthanized by day 21 after challenge, we found that only IFNα-treated *Ifnar1*$^{fl/fl}$ mice showed an increased proportion of Tcm and decreased percentage of Tn when compared to NaCl-treated *Ifnar1*$^{fl/fl}$ controls (*Figure 7C*), suggesting that IFNα-responsive LSECs may promote antitumor immune memory in secondary lymphoid organs.

To assess whether IFNα-stimulated HECs and LSECs promoted memory responses endowed with antitumor potential, *Ifnar1*$^{fl/fl}$-cured mice (defined as animals that 7 days after IFNα treatment initiation were intramesenterically challenged with MC38 cells and survived as disease-free animals until day 50) or naive *Ifnar1*$^{fl/fl}$ control mice were subcutaneously rechallenged with MC38 cells (*Figure 7D*). Notably, while the latter animals developed subcutaneous tumors that increased in size over time, none of the *Ifnar1*$^{fl/fl}$-cured mice showed detectable lesions at any time point studied (*Figure 7E*). These results indicate that continuous IFNα treatment promotes protection against secondary tumor challenge even after IFNα therapy discontinuation. The results also suggest that this effect may be dependent on the capacity of IFN-sensitive HECs and LSECs to foster antitumor immunity, especially tumor-specific effector CD8$^+$ T cell responses that are well-known to control tumor growth in vivo in different experimental settings (*Dobrzanski et al., 2000*; *Katlinski et al., 2017*; *Klebanoff et al., 2005*; *Yu et al., 2019*).

## Discussion

In this study, we used different mouse models of CRC liver metastasis to show that the continuous perioperative administration of relatively low IFNα doses provides significant antitumor potential in vivo without provoking overt toxicity. Moreover, under the pharmacological conditions we defined (route, dosage, treatment duration, and chemical nature of the recombinant protein), we did not observe counter-regulatory mechanisms affecting IFNα efficacy (*Katlinski et al., 2017*), or significant systemic side effects, as our strategy avoids the short tissue-oscillatory IFNα bursts that are

often achieved after high and pulsed administrations, often associated with efficacy-limiting toxicities (*Weber et al., 2015*). These results are consistent with previous preclinical work indicating that the intrahepatic delivery of IFNα through a gene/cell therapy approach curbs CRC liver metastases by acting primarily on unidentified non-hematopoietic stromal cell populations (*Catarinella et al., 2016*).

Given the pleotropic nature of IFNα, we demonstrated that the antimetastatic activity of IFNα is neither based on the direct inhibition of primary intracecal tumor growth, favoring the hypothesis that IFNα therapy does not modify the number of cells that spread from primary tumors and seed into the liver – nor on the direct inhibition of metastatic cell growth within the liver. These data are consistent with the high IFNα concentrations required to activate the 'tunable' direct antiproliferative functions of this cytokine, likely exceeding the levels achieved in our system (*Catarinella et al., 2016*; *Schreiber, 2017*). In addition, IFNα therapy does not require indirect stimulation of hepatocytes, HSCs, DCs, KCs, or LCMs to exert its antimetastatic functions. Rather, the results pinpointed HECs/LSECs as key local and early sensors of IFNα that ultimately limit CRC cell invasion into the liver.

Mechanistically, we showed that IFNα-stimulated LSECs inhibit the trans-sinusoidal migration of circulating CRC cells normally occurring within 24 hr of their initial intrahepatic landing. This effect is associated with phenotypic changes that IFNα-stimulated LSECs acquire or induce in the liver microenvironment. Among these changes, we observed a reduction in the overall LSEC porosity (i.e. sinusoidal fenestrae were reduced in number and size), an enhancement in the subendothelial deposition of basal membrane components (including Collagen IV and Laminin) and an upregulation of LYVE-1, a marker of hepatic capillarization (*Pandey et al., 2020*; *Wohlfeil et al., 2019*). Along these lines, it is noteworthy that in the 'healthy' liver, functioning as a common site for CRC metastases, LSECs contain numerous fenestrae of up to 200 nm in diameter and normally lack the typical basal membrane that characterizes the microvasculature of most other tissues and organs (*Jacobs et al., 2010*). It is also interesting to note that IFNα-stimulated LSECs promote microvascular alterations like those typifying pathological conditions (e.g. initial hepatic capillarization and liver fibrosis *Pandey et al., 2020*; *Wohlfeil et al., 2019*) associated with impaired immune cell extravasation and reduced immune surveillance (*Guidotti et al., 2015*) and reduction of hepatic metastases from solid tumors including CRCs (*Wohlfeil et al., 2019*). This fits with the evidence that CRC patients suffering from chronic viral liver fibrotic diseases characterized by hepatic endogenous type I interferon production display lower incidence of hepatic metastases (*Augustin et al., 2013*; *Baiocchini et al., 2019*; *Li Destri et al., 2013*). The existence that fibrotic liver diseases not associated with reduced metastatic risk (*Kondo et al., 2016*) suggests that changes in the vascular hepatic niche other than matrix deposition play additional roles in this process. Indeed, IFNα stimulated LSEC-governed changes hampering CRC extravasation including the modification of the sinusoidal GCX that, by increasing its thickness and modifying its chemical composition, recapitulated conditions known to negatively regulate the trans-endothelial migration of tumor cells in other settings (*Glinskii et al., 2005*; *Mitchell and King, 2014*; *Offeddu et al., 2021*; *Wilkinson et al., 2020*). The continuous administration of therapeutic low-doses of IFNα thus stimulate HECs/LSECs to shape a vascular antimetastatic barrier preventing the interaction between tumor cells and endothelial cells that are known to promote the extravasation of the former cells (*Glinskii et al., 2005*; *Mitchell and King, 2014*; *Wilkinson et al., 2020*). Accordingly, the enhanced expression of 'pro-migratory' adhesion molecules and integrins that we observed in the liver of animals bearing IFNα-responsive LSECs appear to be efficiently counteracted by the creation of such vascular barrier.

Of note, the functional consequences of LSEC capillarization (especially the induction of hepatic fibrosis) during states of chronic liver injury highly depend on the relative magnitude and duration of the underlying liver disease (*DeLeve, 2015*). Additionally, both LSEC capillarization and hepatic fibrosis are reversed when chronic liver injury resolves (*DeLeve, 2015*). In keeping with these concepts and the absence of sALT elevation or morphological evidence of liver disease during continuous IFNα therapy, it is not surprising that we observed a complete recovery of fenestrae abundance and LSEC porosity 40 days after therapy discontinuation. This supports the notion that a continuous but relatively short IFNα therapy promotes changes in the structure and function of LSECs that are mild and reversible and should not result in persistent hepatic fibrinogenesis. Such notion is also supported by the absence of hepatic toxicity (*Weber et al., 2015*) and the significant reduction in established fibrosis in patients with chronic viral liver diseases treated with recombinant IFNα for up to 48 weeks (*Li et al., 2019*; *Poynard et al., 2002*). The notion that IFNα treatment failed to shape the vascular

antimetastatic barrier in mice carrying the *Ifnar1*-deficiency only in endothelial cells further strengthens this hypothesis and places HECs/LSECs at a center of a relevant antitumor process ultimately limiting CRC liver invasion. This concept is also indirectly supported by the fact that the increased expression *Gata4* -a master-regulator of liver sinusoidal differentiation which leads to liver fibrosis deposition upon its loss (*Winkler et al., 2021*) - and *Smad7* – an inhibitor of transforming growth factor-beta (TGF-β) dependent subendothelial matrix deposition and sinusoidal capillarization (*Tauriello et al., 2018*) were not downregulated by IFNα treatment in mice in which all cell types except LSECs could sense this cytokine, thus loosening the hepatic vascular barrier.

In addition, to hindering the initial trans-sinusoidal migration of CRC cells in vivo, IFNα-stimulated LSECs efficiently cross-presented nominal tumor antigens to naive CD8[+] T cells in vitro, enabling degrees of T cell priming and effector differentiation that were comparable to those induced by professional APCs. In keeping with this, we demonstrated that the in vivo IFNα stimulation of LSECs resulted in the upregulation of proteins and transcripts associated with antigen processing and presentation or co-stimulation (e.g. MHC-I, CD86, IL-6RA, *B2m, Tap1, Psmb-8/9 and H2-d1, H2-k1/2*) (*Böttcher et al., 2014*; *Katz et al., 2004*; *Montoya et al., 2002*; *Rodriguez et al., 1999*). Moreover, our results also suggest that IFNα-stimulated LSECs may play a key role in antitumor immunity, as mice were protected from secondary tumor rechallenge even after discontinuation of IFNα treatment. The fact that the same IFNα therapy also significantly increased the overall number of central memory T cells in the spleen while decreasing that of naive T cells (*Sallusto et al., 2004*; *Yu et al., 2019*) further suggests a role for IFNα-stimulated LSECs in the generation of systemic and protective long-term antitumor immunity.

These data are consistent with the notion that IFNα-stimulated LSECs, due to their anatomical proximity and efficient endocytosis capacity that is among the highest of all cell types in the body (*Sorensen and Smedsrod, 2020*) – rapidly remove CRC-derived antigens from the intravascular space and productively and rapidly contribute to the development of effective antitumor immunity, since this process does not require the time-consuming step of migration to lymphatic tissue (*Böttcher et al., 2014*). This concept is also supported by the upregulation by IFNα-stimulated LSECs of *Cxcl9* and *Cxcl10*, two chemokines involved in the attraction and retention of naïve T cell populations of lymphocytes into the liver (*Franciszkiewicz et al., 2012*), a necessary step for the generation of an efficient antitumor immune response. Additionally, other cell types within the hepatic niche could further amplify this IFNα-initiated cascade, as it has been shown that dendritic cells releasing IFNα also reduce liver metastatic colonization by CRC cells (*Toyoshima et al., 2019*) and that this cytokine properly polarizes the tumor microenvironment (*Catarinella et al., 2016*; *De Palma et al., 2008*). On the contrary, the notion that a minority of IFNα-treated animals develop small intrahepatic lesions that display similar proliferation, neoangiogenic and immunologic markers than untreated lesions high-lights the possibility that CRC tumors, once established as macroscopic metastases, may become refractory and resistant to IFNα therapy by downregulating *Ifnar1* (*Boukhaled et al., 2021*; *Katlinski et al., 2017*). This would be consistent with the lack of efficacy of our approach in established orthot-opic tumors within the cecal wall.

Altogether, we have identified a novel MoA by which IFNα functions as antitumor drug against CRC liver metastases. Whether the adoption of similar LSEC-stimulating IFNα treatments may also curb the hepatic growth of metastatic cells originating from other solid tumors, or if continuous IFNα treatment promote the generation of vascular barriers in other metastasis-prone organs remains to be determined (*Crist and Ghajar, 2021*). Based on the findings of this report, we propose the following model: CRC cells emerging from the primary tumor reach the hepatic sinusoids via the portal circulation and arrest – mostly because of size constrains – at the portal side of the sinusoidal circulation. CRC cells then trans-sinusoidally migrate into the liver parenchyma and develop micrometastases that will eventually grow overtime, promoting the generation of an immunosuppressive microenvironment. Continuous therapy with well-tolerated doses of recombinant IFNα, stimulates HECs/LSECs to limit CRC trans-sinusoidal migration and parenchymal invasion by building up a vascular barrier typified by the reduction of LSECs porosity, the increased thickness of GCX and the appearance of a basal membrane. Continuous IFNα therapy also promotes long-term antitumor immunity in cured mice and protection from secondary tumor challenge, by stimulating LSECs to efficiently cross-prime tumor antigens to naïve CD8[+] T cells (*Figure 7F*).

In terms of future clinical applications, our strategy could be used as perioperative neoadjuvant immunotherapy in CRC patients undergoing resection of their primary tumor who are at high risk for developing metachronous liver metastases (*Engstrand et al., 2019*; *van Gestel et al., 2014*). Indeed, several technologies have already been developed for the sustained release of drugs, such as osmotic pumps, electronic devices, hyaluronic acid-based hydrogels (*Park et al., 2018*; *Stewart et al., 2018*; *Yun and Huang, 2016*), FDA-approved polymer miscellas – such as pegylated (PEG)-IFNα (*Foser et al., 2003*; *Glue et al., 2000*) – and IFNα cell/gene therapy approaches (*Catarinella et al., 2016*), which could quickly translate our results into clinical practice. Of note, the use of clinically approved doses of pegylated-IFNα has shown improved serum stability and clinical efficacy and reduced side effects, with serum IFNα concentrations similar to those achieved in our system (*Foser et al., 2003*; *Glue et al., 2000*).

All in all, the results of this study support the use of continuous low doses of IFNα as an antimetastatic drug during the perioperative period, due to its ability to transform a metastases-prone liver into a metastases-resistant organ.

# Materials and methods

## Animal studies

Eight- to 10-week-old C57BL/6 J and BALB/c mice were purchased from Charles River Laboratory, Calco, Italy. CB6 mice were obtained by crossing *Mus musculus* inbred C57BL/6 J male mice (H-2b restricted) with *Mus musculus* inbred BALB/c female mice (H-2d restricted), to produce H-2bxd F1 hybrids. *Ifnar1*$^{fl/fl}$ mice on a C57BL/6 J background (B6(Cg)-*Ifnar1*tm1.1Ees/J, JAX:028256), *Alb*$^{Cre}$ (B6.Cg-Speer6-ps1Tg(*Alb*-cre)21Mgn/J, JAX:003574), *Pdgfrb*$^{CreERT2}$ (B6.Cg-Tg(Pdgfrb-cre/ERT2)6096Rha/J JAX:029684), *Itgax*$^{Cre}$ (CD11c, B6.Cg-Tg(*Itgax*-cre)1-1Reiz/J, JAX:008068), and Rosa26-ZsGreen (B6.Cg-Gt(ROSA)26Sortm6(CAG-ZsGreen1)Hze/J, JAX:007906) reporter mice were purchased from the Jackson Laboratory. *Cdh5*(PAC)$^{-CreERT2}$ mice (*VeCad*$^{CreERT2}$) (*Wang et al., 2010*) were kindly provided by S. Brunelli (UniMib, Milan). NOD-scid IL2Rγnull (NSG) immunodeficient mice (NOD.Cg-Prkdcscid Il2rgtm1Wjl/SzJ, JAX:005557) and OT-I mice (C57BL/6-Tg(TcraTcrb)1100Mgb/Crl, JAX:003831) were purchased from Charles River Laboratory.

The conditional deletion of *Ifnar1* was obtained by crossing mice carrying loxP-flanked *Ifnar1* (*Prigge et al., 2015*) with transgenic mice expressing Cre recombinase under the control of either endothelial cell (*VeCad*$^{CreERT2}$), stellate cell (*Pdgfrb*$^{CreERT2}$), hepatocyte (*Alb*$^{Cre}$) and dendritic cell (*Itgax*$^{Cre}$) promoters. To induce the Cre recombination and *Ifnar1* deletion into *VeCad*$^{CreERT2}$, Cre recombinase was induced by three subcutaneous injections of Tamoxifen 50 mg/kg at p5, p6 and p7, as previously described (*Tirone et al., 2018*), while *Pdgfrb*$^{CreERT2}$ adult mice were treated by three consecutive ip injections of Tamoxifen 100 mg/kg. Upon treatment, the exon3 of *Ifnar1* is excised resulting in a loss of *Ifnar1* in endothelial and stellate cells. To ensure that *Ifnar1* exon 3 of was efficiently excised hepatic DNA was isolated from the liver of 8-week-old *VeCad*$^{Ifnar1\_KO}$, *Pdgfrb*$^{Ifnar1\_KO}$, *Alb*$^{Ifnar1\_KO}$ and *Itgax*$^{Ifnar1\_KO}$ mice and polymerase chain reaction (PCR) analysis using *Ifnar1* intron 3-forward and *Ifnar1* intron 3-reverse oligonucleotides was performed. To control for possible changes in the microbiota composition of *VeCad*$^{Ifnar1\_KO}$, *Pdgfrb*$^{Ifnar1\_KO}$, *Alb*$^{Ifnar1\_KO}$ and *Itgax*$^{Ifnar1\_KO}$ and *Ifnar1*$^{fl/fl}$ littermates, mice from each litter and cage were randomly allocated into the experimental groups and were co-housed or systematically exposed to beddings of the other groups to ensure the same exposure to the microbiota.

## Study approval

All animal experiments were approved by the Animal Care and Use Committee of the San Raffaele Scientific Institute (SRSI, IACUC 691, 808 and 1042) and were conducted in specific pathogen-free (SPF) facility in microisolator cages under a 12 hr light/dark cycle with free access to water and standard mouse diet (Teklad Global 18% Protein Rodent Diet, Harlan).

## CRC cell lines

CT26 (H-2d, BALB/c-derived) cell line was purchased from ATCC. MC38 (H-2b, C57BL/6-derived), have been previously described (*Catarinella et al., 2016*). MC38 cells were transduced with a PGK-GFP lentiviral vector, cloned and sorted by FACS to establish MC38$^{GFP}$ fluorescently tagged cell lines (*Talamini et al., 2021*). All cells were routinely tested for mycoplasma contamination using the

N-GARDE Mycoplasma PCR reagent set (EuroClone). CT26 and CT26$^{LM3}$ cells were cultured under standard condition at 37 °C in a humid atmosphere with 5% $CO_2$ in RPMI GlutaMAX medium (Gibco) supplemented with 10% FBS (Lonza) and 1% penicillin/streptomycin (P/S) (Gibco). MC38, MC38$^{GFP}$, MC38$^{Ifnar1\_KO}$ and cells were cultured under standard condition at 37 °C in a humid atmosphere with 5% $CO_2$ in DMEM GlutaMAX medium (Gibco) supplemented with 10% FBS (Lonza) and 1% P/S (Gibco). Cell number and dimension was routinely assessed by automated CytoSMART cell counter (Corning). Details of murine cell lines used in the experiments (source of cell lines, background and origin of cancer) are mentioned in key resource table. The tumor cell lines MC38, MC38$^{GFP}$, MC38$^{Ifnar1\_KO}$ and CT26 were passaged in vivo once before use in experiments.

## Mouse models of liver metastases

Eight-to ten-week-old sex- and age-matched mice were injected with $5 \times 10^3$ CT26 or $5 \times 10^4$ MC38 CRC cell lines either through intrasplenic or superior mesenteric vein injections as previously described (*Catarinella et al., 2016*; *van der Bij et al., 2010*). For early time point experiments, $7 \times 10^5$ MC38$^{GFP}$ cells were injected in the superior mesenteric vein of anesthetized mice as described (*van der Bij et al., 2010*). For intrasplenic or superior mesenteric vein injections, deep anesthesia was induced by isoflurane inhalation (5% induction and 2% for maintenance in 2 l/min oxygen). The indicated number of CRC cells was injected into spleen or the superior mesenteric vein using a 29 G needle and to prevent excessive bleeding vein puncture was compressed with a sterile and absorbable hemostatic gauze (TABOTAMP). The peritoneum and skin were sutured with silk 4.0 and 7 mm wound clips as described (*Catarinella et al., 2016*). This experimental setting may mimic the vascular spreading of CRC cells during primary tumor resection and, thus, preventive IFNα infusion may be considered as a neoadjuvant treatment.

## Mouse model of orthotopic colorectal cancer liver metastases

The generation of highly metastatic CT26 CRC cells was obtained by three consecutive rounds of in vivo selection as previously reported (*Zhang et al., 2013*). Briefly, $2 \times 10^6$ CT26 cells were first injected subcutaneously into the right flank of immunodeficient NSG mice. After 28 days, tumors were excised, dissected, sliced into small fragments, and digested for 30 min at 37 °C in DMEM containing collagenase type IV (200 units/ml; Sigma-Aldrich) and Dnase I (100 units/ml; Sigma-Aldrich). The resulting cell suspension defined as CT26$^{sc}$, were maintained at 4 °C, filtered through a 70 μm nylon cell strainer (BD Biosciences, Bedford, MA), washed in PBS, and grown in RPMI 10% FBS. Sub-confluent CT26$^{sc}$ cells were harvested, resuspended in PBS:Matrigel (1:1) (Corning, MERK) and then injected into the cecal wall of immune competent anesthetized (isoflurane, 5% induction and 2% for maintenance in 2 l/min oxygen), CB6 recipient mice as described (*Zhang et al., 2013*). Briefly, a midline incision was made to exteriorize the cecum. Using a 33 G micro-injector (Hamilton, USA), 10 μl of a 50% Matrigel solution (BD Bioscience, USA) containing $2 \times 10^5$ CT26$^{sc}$ cells were injected into the cecum wall. To avoid intraperitoneal spreading of CT26$^{sc}$, the injection site was sealed with tissue adhesive (3 M Vetbond, USA) and washed with 70% alcohol. The cecum was replaced in the peritoneal cavity, and the abdominal wall and skin incision was sutured with silk 4.0 and 7 mm wound clips as described (*Catarinella et al., 2016*). Twenty-eight days after injection, mice were euthanized and single cell suspensions of liver metastatic lesions, defined as CT26$^{LM1}$, were obtained as described above. This cycle was repeated twice to obtain the highly metastatic CT26$^{LM3}$ cells.

## Recombinant mouse IFNα therapy

Continuous intraperitoneal IFNα delivery (IFNα1 carrier-free, Biolegend, San Diego, CA, USA) was achieved by intraperitoneal implantation of mini-osmotic pumps (MOP, ALZET, Cupertino, CA, USA) able to deliver either 50, 150, or 1050 ng IFNα a day for 14 or 28 days. NaCl-containing MOP were used as controls. Within each specific experiment, mice of each genotype were randomly assigned to receive either NaCl- or IFNα-containing MOP. MOP filling, priming and implantation within the peritoneum was performed following manufacturer's instructions. To avoid MRI artifacts due to the presence of metallic components within MOP, the day before MRI acquisition, MOP were removed from the peritoneum. To directly investigate responsiveness of liver cells to IFNα, signaling downstream of *Ifnar1* receptor was assessed by measuring pSTAT1 by IHC 30 min after an ip injection of NaCl or 1 μg IFNα, a dose able to synchronize pSTAT1 expression in all *Ifnar1* expressing cells (*Lin et al., 2016*).

## Tumor rechallenge of IFNα cured mice

IFNα-cured mice that were designated as MC38-tumor free for at least 50 days after challenge, were subcutaneously rechallenged with $5x10^3$ MC38 cells resuspended in 200 µl of PBS:Matrigel (1:1). Age-matched naïve syngeneic mice were used as control. Tumor volumes were measured twice a week and euthanized for ethical reasons when tumor size reached ~500 mm$^3$.

## Magnetic resonance imaging (MRI)

All MRI studies were carried out at the Experimental Imaging Center of SRSI on a preclinical 7-Tesla MR scanner (Bruker, BioSpec 70/30 USR, Paravision 6.0.1, Germany) equipped with 450/675 mT/m gradients (slew rate: 3400/4500 T/m/s; rise time 140 µs), coupled with a dedicated 4 channels volumetric mouse body coil. All images were acquired in vivo, under inhalational anesthesia (Isoflurane, 3% for induction and 2% for maintenance in 1 L/min oxygen) with mice laid prone on the imaging table. A dedicated temperature control system was used to prevent hypothermia; respiratory rate and body temperature were continuously monitored (SA Instruments, Inc, Stony Brook, NY, USA) during the whole MRI scan. An intravenous injection of gadoxetic acid (Gd-EOB-DTPA; Primovist, Bayer Schering Pharma) at a dose of 0.05 µmol/g of body weight was administered via the tail vein before placing the mice on the scanner table. As previously described (*Sitia et al., 2012*), the MRI studies relied on an axial fat-saturated T2-weighted sequence (TurboRARE-T2: TR = 3394ms, TE = 33ms, voxel-size=0.125 × 0.09 x0.8mm, averages = 3) acquired immediately after Gd-EOB-DTPA injection and an axial fat-saturated T1-weighted scan (RARE-T1: TR = 581ms, TE = 8.6ms, voxel-size=0.125 × 0.07 x0.8mm, averages = 4) acquired thereafter, during the hepatobiliary phase (HBP) of contrast excretion (starting from 10 min after Gd-EOB-DTPA injection). Two board certified radiologists skilled in clinical and preclinical abdominal MR imaging, blinded to any other information, reviewed all MRI studies using an open-source image visualization and quantification software (Mipav, 5.3.4 and later versions, Biomedical Imaging Research Services Section, ISL, CIT, National Institute of Health, USA). Liver metastases were identified as focal lesions showing slight hyper-intensity on T2-weighted images and concurrent hypo-intensity on contrast-enhanced HBP T1-weighted images. Liver metastases segmentation was performed by manual drawing of regions-of-interest (ROIs) on each slice, yielding volumes-of-interest (VOIs; lesion area x slice thickness) for the entire sequence. The total CRC metastatic mass was obtained by summing up the volumes of all single VOIs that were semi-automatically provided by the software.

## MC38 gene editing

To knockout *Ifnar1* in MC38 cell line, we used GeneArt CRISPR Nuclease Vector with OFP Reporter Kit (ThermoFisher Scientific). Target-specific CRISPR RNA guides (crRNA) were designed using GeneArt CRISPR Search and Design Tool (ThermoFisher Scientific) and the two following crRNAs with the fewest predicted off-target effects were selected: crRNA1: TAGACGTCTATATTCTCAGGGTTTT; crRNA2: ATGTAGACGTCTATATTCTCGTTTT. Annealed crRNAs were cloned in GeneArt CRISPR Nuclease Vector with OFP according to the manufacturer's instructions and vector constructs were used to transform OneShot TOP10 chemically competent *E. coli* cells (ThermoFisher Scientific). Vectors were validated by Sanger sequencing of DNA from 10 colonies for each crRNA (LightRun GATC Carlo Erba). To express the CRISPR–Cas9 system transiently, 2 µg of each vector were used to transfect $5x10^5$ MC38 cell line with Lipofectamine 2000 (ThermoFisher Scientific) in Opti-MEM medium (ThermoFisher Scientific). After 3 days, transfection efficiency was evaluated measuring the orange fluorescence protein (OFP) by FACS (LSRFortessa) and data analysed using FlowJo v10.5. OFP positive bulk populations were single-cell cloned in 96-well plates and a total of 12 clones screened for mismatches by T7E1 assay (New England Biolabs). Briefly, genomic DNA from MC38 clones was extracted by Qiamp mini kit (Qiagen) and amplified within the exon 2 of *Ifnar1* locus using the following oligos: *Ifnar1* exon2 forward, TCCAAGACTCCTGCTGTC and *Ifnar1* exon2 reverse: GCACTTTTACTTGCTC GGT. The PCR products were denatured and annealed according to manufacture protocol (New England Biolabs). The digestion reaction was run onto 2% agarose gel to identify mismatched clones. Genetic validation of T7E1-positive clones was assessed by cloning the PCR products with TOPO TA Cloning, Dual Promoter, Kit (ThermoFisher Scientific) and subsequently 10 clones per cell line were sequenced (LightRun GATC Carlo Erba). Functional validation of *Ifnar1* knockout MC38 cells was

determined by RT-PCR for the ISG *Irf7* after 4 hr of in vitro stimulation with 300 ng/ml IFNα, using MC38 transfected non-edited WT clone as control (MC38).

## Peripheral blood analyses

At the indicated time points after IFNα administration, whole anti-coagulated blood of MOP-NaCl and MOP-IFNα-treated mice was collected from the retro-orbital plexus of anesthetized animals (isoflurane, 5% for induction and 2% for maintenance in 2 l/min oxygen) using Na-heparin coated capillaries (Hirschmann Laborgeräte GmbH, Germany) and vials (Microvette, Sarstedt, Germany). Hematologic parameters were evaluated using an automated cell counter (ProCyte Dx, IDEXX Laboratories, USA). The extent of hepatocellular injury was monitored by measuring serum ALT (sALT) activity at several time points after IFNα treatment, as previously described (*Sitia et al., 2012*).

## Measurement of plasma IFNα by ELISA

Circulating levels of IFNα were quantified in plasma collected from NaCl controls or IFNα-treated mice at indicated time points using VeriKine-HS Mouse IFNα all Subtype ELISA Kit (PBL) according to the manufacturer's instructions. The IFNα titer in the samples was determine by plotting the optical density (OD) subtracted of blank OD to eliminate background, using a 4-parameter logistic fit for the standard curve by using Prism v8 (GraphPad). Detection range is comprised between 2.38 and 152 pg/ml of IFNα.

## RNA extraction and quantitative real-time PCR gene expression analyses

Total RNA was isolated from liver homogenates of MOP-NaCl control and MOP-IFNα-treated mice by using the ReliaPrep RNA Tissue Miniprep System (Promega) and DNAse TURBO (Thermo Fisher Scientific) following manufacturer's recommendation. The extracted RNA was subsequently retrotranscribed to cDNA as previously described (*Sitia et al., 2011*). Quantitative real-time PCR analysis was performed utilizing the ViiA7 Fast Real-Time PCR System (Applied Biosystems). The ISG, *Irf7* (Mm00516793_g1) as well as the housekeeping *Gapdh* (Mm 99999915_g1) were quantified by using the indicated FAM-MGB labeled TaqMan Gene Expression Assays (Applied Biosystems). Gene expression was determined as the difference between the threshold cycle (Ct) of the gene of interest (Goi) and the Ct of the housekeeping (*Gapdh*) of the same sample (ΔCt). The fold-change expression of each Goi was calculated over its basal expression in the control sample by the formula $2^{-\Delta\Delta Ct}$ as described (*Sitia et al., 2012*).

## Immunohistochemistry

At time of autopsy for each mouse, livers were perfused with PBS, harvested and different pieces were sampled, fixed in zinc-formalin, processed and embedded in paraffin for histological and immunohistochemical analysis, as previously described (*Sitia et al., 2012*). Immunohistochemical staining using a Bond RX Automated Immunohistochemistry (Leica Microsystems GmbH, Wetzlar, Germany) was performed on 3-μm-thick sections. First, tissues were deparaffinized and pre-treated with the Epitope Retrieval Solution [ER1 Citrate Buffer for Ki-67 (dilution 1:200, clone SP6, Thermo Fisher Scientific) and F4/80 (dilution 1:200, clone A3-1, Bio-Rad); ER2 EDTA for CD3 (dilution 1:100, clone SP7, Abcam),CD34 (dilution 1:300, clone MEC14.7, Biolegend) and pSTAT1 (dilution 1:800, clone 58D6, Cell Signaling)] at 100 °C for 30 min. After washing steps, peroxidase blocking was carried out for 10 min using the Bond Polymer Refine Detection Kit DS9800 (Leica Microsystems GmbH). Then, tissues were washed and incubated for 1 hr RT with the primary antibody diluted in IHC Antibody Diluent (Novocastra, Leica RE7133). Subsequently, tissues were incubated with polymer-HRP or Rat-on-Mouse HRP (Biocare Medical, RT517H), developed with DAB-Chromogen for 10 min and counterstained with Hematoxilin for 5 min. For image acquisition and analysis, eSlide Manager (Aperio Leica Biosystems) was used. All images were acquired using the Aperio AT2 system (Leica Biosystems). Quantifications were performed by automated image analysis software through dedicated macros of the ImageScope program, customized following manufacturer's instructions (Leica Biosystems). The images shown were identified as representative area of interest within the total area of the specimen analyzed and exported as ImageScope snapshots.

## Immunofluorescence and confocal microscopy

Livers were perfused with PBS, harvested and fixed over-night in 4% paraformaldehyde (PFA), equilibrated in 30% sucrose in PBS over-night at 4 °C prior to embedding in OCT (Bio-Optica) for quick freezing at –80 °C. Thirty-µm-thick cryosections were adhered to Superfrost Plus slides (Thermo Scientific). For immunofluorescence staining, sections were blocked and permeabilized with PBS containing 5% FBS and 0.1% Triton X-100 (Sigma-Aldrich) for 30 min at room temperature and subsequently incubated with 10% of donkey serum (DS; Sigma-Aldrich) in PBS for 30–60 min at room temperature. Staining with primary and secondary antibodies, were performed with staining buffer (PBS with 1.5% DS, 0.2% Triton X-100 and 1% BSA), using the following antibodies and dilutions: anti-PDGFRβ/CD140b (dilution 1:200, clone APB5, eBioscience)+anti rat AF647 (dilution 1:200, Jackson IR); anti-CD11c AF647, (dilution 1:100, clone N418, Biolegend); anti-GFP (dilution 1:100, rabbit polyclonal, A11122 Thermo Fisher Scientific)+anti-rabbit AF 488 (dilution 1:200, Thermo Fisher Scientific); anti-CD31/PECAM-1 (dilution 1:300, goat polyclonal, AF3628 R&D Systems)+anti goat AF546 (dilution 1:200, Thermo Fisher Scientific); anti-Heparan Sulfate (dilution 1:50, clone F58-10E4, Amsbio)+anti IgM conjugated to APC (dilution 100, clone II/41, Thermo Fisher Scientific); anti-LYVE-1 (dilution 1:300, rabbit polyclonal, Novus Biologicals)+anti-rabbit AF647 (dilution 1:200, Thermo Fisher Scientific); anti-Collagen type IV (dilution 1:100, rabbit polyclonal, Abcam)+anti-rabbit AF488 (dilution 1:200, Thermo Fisher Scientific); anti-Laminin (dilution 1:300, rabbit polyclonal, Sigma-Aldrich)+anti-rabbit AF488 (dilution 1:200, Thermo Fisher Scientific); anti-CD54/ICAM1 (dilution 1:100, clone YN1/1.7.4, Biolegend)+anti rat AF647 (dilution 1:200, Jackson IR); anti-CD62E/E-selectin (dilution 1:100, clone 10E9.6, BD Bioscience)+anti rat AF647 (dilution 1:200, Jackson IR). Confocal images were acquired using a Leica SP8 confocal systems (Leica Microsystems) that are available at the SRSI Advanced Light and Electron Microscopy BioImaging Center (ALEMBIC). Fifteen–20 µm z-stacks were projected in 2D and processed using Fiji image processing software (*Schindelin et al., 2012*). Localization of MC38[GFP] tumor cells within liver vessels, 20–30 square xy sections (1024x1,024 pixel) confocal xyz stacks, from NaCl- and IFNα-treated mice, were acquired with 0.5 µm z-spacing on a Leica TCS SP8. The Imaris Surpass View and Surface Creation Wizard were used to create 3D renderings of MC38[GFP] cells and CD31[+] liver vessels as previously reported (*Guidotti et al., 2015*). A tumor cell was considered intravascular when at least 95% of its surface-reconstructed body was inside the vessel lumen in all the acquired sections projected in the horizontal (xy), transversal (yz) and longitudinal (xz) planes. Entire liver sections were acquired using MAVIG RS-G4 confocal microscope (MAVIG GmbH Research, Germany) to quantify the number of MC38[GFP] tumor cells in relation to the total liver area. The quantification of the percentage of liver area covered by endothelial markers and extracellular matrix components, such as Heparan Sulfate, Laminin, Collagen type IV, ICAM1 and E-Selectin, was evaluated using ImageJ software, applying the same threshold to the different experimental groups for each channel and measuring the percentage of area limited to threshold. Colocalization analysis of GFP in the different Cre recombinant mouse models was performed using an unsupervised ImageJ plugin algorithm termed Colocalization, which was developed by Pierre Bourdoncle (Institut Jacques Monod, Service Imagerie, Paris; 2003–2004).

## Scanning and transmission electron microscopy

Electron microscopy (EM) fixative composition was 2,5% glutaraldehyde, 2% paraformaldehyde, 2 mM $CaCl_2$ and 2% sucrose in 0.1 M Na cacodylate buffer (pH 7,4). For the analysis of endothelial GCX, the EM fixative was supplemented with 2% Lanthanum nitrate (MERK) as previously reported (*Inagawa et al., 2018*). Warm PBS and EM fixative at 35–37°C was used to ensure tissue integrity. When mice were under deep anesthesia, with a single ip injection of 50–60 mg/kg Tribromoethanol (Avertin), a Y incision was made in the abdomen to expose the liver and the portal vein. The portal vein was cannulated with an appropriately sized IV cannula of 22 G and the liver was perfused with 15 ml of warm PBS at a constant rate of 3 rpm using a peristaltic pump (Peri-Star Pro, 2Biological Instruments) as previously reported (*Guidotti et al., 2015*). In situ fixation was achieved by perfusing EM fixative for approximately 5 min at 3 rpm. Fixed liver was harvested and cut into 5 mm blocks using a scalpel. Liver blocks were ulteriorly immersed in EM fixative for 24–72 hr at 4 °C and finally EM fixative was replaced with 0.1 M sodium cacodylate buffer and stored at 4 °C until processed for TEM or SEM analysis. Liver blocks were post-fixed in 1% osmium tetroxide ($OsO_4$), 1,5% potassium ferricyanide($K_4[Fe(CN)_6]$) in 0.1 M Na Cacodylate buffer for 1 hr on ice. Afterwards, for SEM, 150-µm-thick

sections were obtained from perfused livers using a vibratome (Leica VT1000S). Sections were further post-fixed in 1% OsO$_4$ in 0.1 M Na Cacodylate, dehydrated through a series of increasing concentration of ethanol and immersed in absolute hexamethyldisilazane (HMDS) that was left to evaporate overnight. Dried sections were sputter-coated with Chromium using a Quorum Q150T ES sputter coater. Sections were then mounted on SEM stubs using conductive adhesive tape and observed in a field-emission scanning electron microscope Gemini 500 (ZEISS, Oberkochen, Germany). The LSEC fenestra measurements were performed from SEM microphotographs taken under a magnification of 20,000 X, using three independent samples from each experimental condition [NaCl-*Ifnar1*$^{fl/fl}$ (n=3), IFNα-*Ifnar1*$^{fl/fl}$ (n=3), NaCl-*VeCad*$^{Ifnar1\_KO}$ (n=3) and IFNα-*VeCad*$^{Ifnar1\_KO}$ mice (n=3)] and a total area of about 720 µm$^2$ of sinusoids was analyzed for each mouse. After 40 days of IFNα discontinuation, two randomly selected liver micrographs of IFNα-*Ifnar1*$^{fl/fl}$ (n=3) were analyzed to determine LSEC fenestra measurements and a total area of approximately 300 µm$^2$ of sinusoidal surface was analyzed. All measurements were made using the ImageJ software, as previously described (*Cogger et al., 2015*). Briefly, the flattened area of the liver sinusoid was selected and longest diameter of each fenestrae was measured. Gaps larger than 250 nm were excluded from the analysis. The average fenestration diameter (defined as the average of all fenestrae diameters excluding gaps area), the fenestration area ($\pi r2$, where r, the radius, was calculated from the individual fenestrae diameter $r=d/2$, without gaps area), the porosity [ $\Sigma (\pi r2)$/(total area analyzed – $\Sigma$(area of gaps))×100; expressed as a percentage], and the fenestration frequency [total number of fenestrations/(total sinusoidal area analyzed – $\Sigma$(area of gaps)); expressed as µm2] have been calculated. Surface roughness analysis of endothelial GCX was determined using Image J software. The flattened area of the liver sinusoid was analyzed, with the SurfChartJ1Q plugin to determine the roughness deviation of all points from a plane after background subtraction, known as Ra coefficient, as previously reported (*Pavlović et al., 2012*). A representative area within the flatten liver sinusoidal surface was used to generate a 3D surface plot image.

For TEM, tissue pieces were rinsed in Na Cacodylate buffer, washed with distilled water (dH2O) and en bloc stained with 0.5% uranyl acetate in dH$_2$O overnight at 4 °C in the dark. Finally, samples were rinsed in dH$_2$O, dehydrated with increasing concentrations of ethanol, embedded in Epon resin and cured in an oven at 60 °C for 48 hr. Ultrathin sections (70–90 nm) were obtained using an ultramicrotome (UC7, Leica microsystem, Vienna, Austria), collected, stained with uranyl acetate and Sato's lead solutions, and observed in a Transmission Electron Microscope Talos L120C (FEI, Thermo Fisher Scientific) operating at 120kV. Images were acquired with a Ceta CCD camera (FEI, Thermo Fisher Scientific). TEM microphotographs were taken under a magnification of 3.400 X, using three independent samples from each experimental condition and a total area of approximately 2.575 µm$^2$ was analyzed for each mouse. LSECs thickness and the width of the space of Disse were measured using ImageJ software. For collagen deposition analysis, at least 10 randomly selected sinusoids from each mouse were analyzed as previously reported (*Gissen and Arias, 2015*; *Warren et al., 2007*).

## Isolation of liver non-parenchymal cells (NPCs)

Liver NPCs, including leukocytes, were isolated from NaCl control or IFNα-treated mice 7 days after MOP implantation, as previously described (*Bénéchet et al., 2019*). Briefly, after euthanasia, the liver was perfused through the vena cava with 5–8 ml of PBS to remove most blood cells. Livers were weighted and 50% of the tissue was sliced in small pieces and incubated 30 min at 37 °C in 10 ml of digestion medium (RPMI GlutaMAX medium [Gibco] containing 200 U/ml of collagenase type IV [Sigma-Aldrich] and 100 U/ml of DNAse I [Sigma-Aldrich]). Remaining undigested fragments were syringed with an 18 G needle and filtered through a 70 µm cell strainer to obtain a single cell suspension. Cells were centrifuged 3 min at 50 g at room temperature and the pellet containing hepatocytes was discarded. The resultant cell suspension of NPCs was incubated for 30 s with ACK lysis buffer (Lonza) to deplete red blood cells, washed with cold RPMI. NPCs were counted and processed for flow cytometry analysis or sorting.

## Isolation of splenocytes and naïve CD8$^+$ T cells

Spleens were obtained from *Ifnar1*$^{fl/fl}$ and *VeCad*$^{Ifnar1\_KO}$ mice 21 days after MC38 cells challenge and placed in a 70 µm cell strainer on a petri dish containing 10 ml of plain RPMI, ground with a syringe plunger to obtain a cell suspension and washed three times with plain RPMI as previously described (*Sitia et al., 2012*). The cell suspension was centrifuged at 400 g for 5 min at room temperature, the

resuspended pellet was incubated for 30 s with ACK lysis buffer and neutralized with RPMI. Resultant splenocytes were counted and processed for FACS analysis. Splenocytes derived from OT1 mice were prepared as described above and naïve CD8$^+$ T cells were isolated using EasySep Mouse Naïve CD8$^+$ T cell isolation kit (STEMCELL Technologies) following manufacturer's instructions.

## Flow cytometry and cell sorting

Cells were resuspended in PBS and LIVE/DEAD Fixable Near-IR or Green dead cell dyes (Thermo Fisher Scientific) and incubated 15 min at RT in the dark for cell viability assessment. Subsequently, cells were blocked with FACS buffer (PBS supplemented with 2% FBS) containing InVivo Mab anti-mouse CD16/CD32 (BioXCell) and stained for surface markers using the following antibodies: anti-CD45R/B220 (clone RA3-6B2, Biolegend), anti-CD11b (clone M1/70, Biolegend), anti-mouse CD11c (clone N418, Biolegend), anti-CD3 (clone 145–2 C11, Thermo Fisher Scientific), anti-CD45 (clone 30-F11, Bioegend), anti-CD8a (clone 53–6.7, Biolegend), anti-F4/80 (clone BM8, Biolegend), anti-CD126/IL-6RA (clone D7715A7, Biolegend), anti-CD18/ITGB2 (clone M18/2, Biolegend), anti-CD49d/ITGA4 (clone 9C10(MFR4.B), Biolegend), anti-H-2Kb/H-2Db (clone 28-8-6, Biolegend), anti-CD86 (clone GL-1, Biolegend), anti-CD146 (clone ME-9F1, Biolegend), anti-CD44 (clone IM7, Biolegend), anti-CD62L (clone MEL-14, Biolegend), anti-IFNγ (clone XMG1.2, BD Biosciences), anti-CD31 (clone MEC13.3, BD Biosciences)for 20 min at 4 °C. For intracellular IFNγ staining, cells were then fixed, permeabilized and stained following Foxp3/Transcription Factor Staining buffer set (Thermo Fisher Scientific) manufacturer's guidelines. When preparing samples for FACS sorting, NPCs were directly blocked and stained with CD45-APC and CD31-BV421. Viability was evaluated by 7-AAD (Biolegend) staining that was added to samples 5 min before sorting. Cell sorting was performed on a BD FACSAria Fusion (BD Biosciences) equipped with four lasers: Blue (488 nm), Yellow/Green (561 nm), Red (640 nm) and Violet (405 nm). 85 µm nozzle was used and sheath fluid pressure was at 45 psi. A highly pure sorting modality (four-way purity sorting) was chosen. The drop delay was determined using BD FACS AccuDrop beads. Unstained and a single-stained controls have been used to set up compensation. Rainbow beads (SPHEROTM Rainbow Calibration Particles) were used to standardize the experiment and were run before each acquisition. Samples were sorted at 4 °C to slow down metabolic activities. Sorted cells were collected in 1.5 ml Eppendorf tubes containing 200 µl of DMEM 10% FBS medium and immediately processed for RNA extraction using ReliaPrep RNA Cell Miniprep System (Promega) and DNAse TURBO (Thermo Fisher Scientific) following manufacturer's recommendation.

## Isolation of HECs, including LSECs and splenic DCs for in vitro studies

Mouse liver perfusion was performed as described in the section Electron Microscopy. After PBS perfusion, liver digestion was achieved in situ by perfusing warm digestion medium at 4 rpm for 10 min. The cava vein was squeezed tight several times to build up some pressure within the liver in order to fill all liver lobes with digestion medium. The resultant digested liver was excised, placed on a petri dish containing digestion medium and the Glisson's capsule was removed. Disaggregated tissue was filtered using a 70-µm cell strainer and centrifuged at 50 g for 3 min to discard hepatocytes. The supernatant containing NPCs was counted and Kupffer cells (KC) were removed using anti-F4/80 Ultrapure microbeads (Miltenyi Biotec) following manufacturer's guidelines. The flow-through of unlabeled NPCs was placed on top of a 25–50% Percoll gradient and centrifuged at 850 g for 20 min at RT without brake and the LSECs located at the 25/50% interface were collected. LSECs were counted and 10$^5$ cells were seeded in a 48-well plate and cultured in collagenated plates with EGM-2 medium (Lonza) for 3 days. For the isolation of splenic DC, spleens were slowly injected with 1 ml of digestion medium until they changed from dark maroon to reddish-orange color. Then, the spleen was minced and pipetted vigorously several times in digestion medium. The cell suspension was filtered using a 70 µm cell strainer, and larger undigested fragments were ulteriorly incubated with digestion medium at 37 °C for 30 min. After tissue digestion, splenocytes were centrifuged at 500 g for 5 min and washed three times with plain RPMI supplemented with 5 mM EDTA to disrupt DC-T cell complexes. Red blood cells were lysed with ACK lysis buffer and DCs cells were isolated using anti-CD11c microbeads UltraPure (Miltenyi) according to manufacturer's instructions. CD11c$^+$ DCs were counted and seeded at 10$^5$ cells in a 48-well plate using RPMI GlutaMAX supplemented with 10% FBS, 1% P/S, 1 x Na Pyruvate (Gibco), 1 x MEM Non-essential Amino Acid Solution (Gibco), 20 µM

β-mercaptoethanol and 40 ng/ml GM-CSF (BioXcell). Every 48 hr 200 µl of fresh medium were added to cultured cells and DCs were grown for 7 days to stimulate DCs differentiation.

## Antigen cross-priming

Prior to naïve CD8[+] T cell co-culture, HECs, including LSECs, and sDCs were stimulated for 18 hr with 1 µg/ml of SIINFEKL peptide (OVA 257–264, Proimmune) or with 1 mg/ml of low endotoxin soluble ovalbumin protein (sOVA; Sigma-Aldrich) in combination with NaCl or 5 ng/ml of IFNα. Subsequently, after extensive washes, cells were co-culture with 5x10[5] naïve CD8[+]T cells isolated from OT-I mice in complete RPMI (containing 10% FBS and 50 µM β-mercaptoethanol) in combination with NaCl or 5 ng/ml of IFNα. After 3 days, CD8[+] T cells were stimulated for 4 hr with 1 µg/ml of SIINFEKL peptide, 5 µg/ml of Brefeldin A (Sigma-Aldrich) and 2.5% EL-4 supernatant in complete RPMI. Finally, the production of IFNγ and the expression of activation markers, such as CD44, were measured by FACS.

## RNA-seq and bioinformatic data analysis

RNA integrity, isolated from sorted liver CD31[+] cells, was evaluated using the Agilent 4100 TapeStation (Agilent Technologies) and samples with RNA integrity number (RIN) above 7.0 were used for subsequent RNA-Seq-based profiling. Libraries were prepared using the Illumina Stranded mRNA ligation kit, according to the manufacturer's instructions. The RNA-Seq library was generated following the standard Illumina RNA-Seq protocol and sequenced on an Illumina NovaSeq 6000 machine (Illumina, San Diego, CA) obtaining an average of 40 millions of single-end reads per sample. The raw reads produced from sequencing were trimmed using Trimmomatic, version 0.32, to remove adapters and to exclude low-quality reads from the analysis. The remaining reads were then aligned to the murine genome GRCm38 using STAR, version 2.5.3 a. Reads were eventually assigned to the corresponding genomic features using featureCounts, according to the Gencode basic annotations (Gencode version M22). Quality of sequencing and alignment was assessed using FastQC, RseQC, and MultiQC tools. Expressed genes were defined as those genes showing at least 1 Count Per Million reads (CPM) on at least a selected number of samples, depending on the size of the compared groups (*Chen et al., 2016*). Low-expressed genes that did not match these criteria were excluded from the corresponding dataset. Gene expression read counts were exported and analyzed in R environment (v. 4.0.3) to identify differentially expressed genes (DEGs), using the DESeq2 Bioconductor library (v. 1.30.1, *Love et al., 2014*). P-values were adjusted using a threshold for false discovery rate (FDR)<0.05 (*Benjamini and Hochberg, 1995*). Significant genes were identified as those genes showing FDR <0.05. Functional enrichment analysis was conducted using the enrichR R package (v. 3.0, *Kuleshov et al., 2016*), starting from the lists of differentially expressed genes as defined by FDR <0.05. Selected pathways were grouped and summarized according to their general biological functions and the hypergeometric test was performed to test the enrichment of these custom genesets, exploiting the hypeR R package (v. 1.8.0). Pre-ranked Gene Set Enrichment Analysis (GSEA *Subramanian et al., 2005*) was performed for each DGE comparison, on all the expressed genes ranked for Log2 fold change. The gene-sets included in the GSEA analyses were obtained from Canonical Pathways, Hallmark and the Gene Ontology collections as they are reported in the MsigDB database.

## Statistical analysis

In all experiments, values are expressed as mean values ± SEM. Statistical significance was estimated by two-tailed non-parametric Mann-Whitney test (e.g. to evaluate differences generated as a consequence of tumor growth) or by one-way ANOVA with Tukey's multiple comparison test when more than two groups were analyzed. Contingency tables were tested by two-tailed Fisher's exact test and Chi-square test. Statistical significance of survival experiments was calculated by log-rank/Mantel-Cox test. All statistical analyses were performed with Prism 8 (GraphPad Software) and were reported in Figure legends. p-values <0.05 were considered statistically significant and reported on graphs. If not mentioned, differences were not statistically significant.

## Acknowledgements

The authors thank B Mendicino, C.B. Ekalle-Soppo, E Daoud, P Giangregorio, M Nicoloso, M Raso, A Fiocchi, P Marra, T Canu, D Lazarevic, E Capitolo, S Bianchessi, V Berno, A Raimondi, A Loffreda, T Catelani, M Mainetti and P Di Lucia for technical support; M Silva for secretarial assistance, C

Tacchetti, P Dellabona and all the San Raffaele AIRC 5X1000 team for helpful discussion. SV Kotenko for help with pSTAT1 IHC protocol. Confocal immunofluorescence histology and electron microscopy was carried out at Alembic, an advanced microscopy laboratory established by the San Raffaele Scientific Institute and the Vita-Salute San Raffaele University. The use of Imaris software was provided by M Iannacone. Acquisition of SEM images was carried out at the Interdepartmental Microscopy Platform of the University of Milan Bicocca. Flow cytometry was carried out at FRACTAL, a flow cytometry resource and advanced cytometry technical applications laboratory established by the San Raffaele Scientific Institute. FRACTAL is an ISO 9001 certified facility and as such all the instruments and equipment are subject to a strict maintenance and functionality check plan. NLT is the recipient of a Swiss National Science Foundation Fellowship P2GEP3_171976; VB is the recipient of a Fondazione Umberto Veronesi "Post-doctoral Fellowships 2019"; AM is the recipient of a Francesco Fronzaroli Memorial fellowship and conducted this study as partial fulfilment of his PhD in Molecular Medicine within the program in Basic and Applied Immunology and Oncology at Vita-Salute San Raffaele University. LGG is supported by the Italian Association for Cancer Research (AIRC) Grant 22737, Lombardy Open Innovation Grant 229452, PRIN Grant 2017MPCWPY from the Italian Ministry of Education, University and Research, Funded Research Agreements from Gilead Sciences and Avalia Therapeutics; GS is supported by Italian Association for Cancer Research (AIRC) grants 18468, 22820 and AIRC 5 per Mille, grant 22737.

## Additional information

### Competing interests

Luca G Guidotti: LGG is a member of the board of directors at Genenta Science and Epsilon Bio and participates in advisory boards/consultancies for Gilead Sciences, Roche, Arbutus Biopharma and Antios Therapeutics. The other authors declare that no competing interests exist.

### Funding

| Funder | Grant reference number | Author |
|---|---|---|
| Associazione Italiana per la Ricerca sul Cancro | 22737 | Luca G Guidotti Giovanni Sitia |
| Associazione Italiana per la Ricerca sul Cancro | 18468 | Giovanni Sitia |
| Associazione Italiana per la Ricerca sul Cancro | 22820 | Giovanni Sitia |
| Regione Lombardia | 229452 | Luca G Guidotti |
| Ministero dell'Istruzione, dell'Università e della Ricerca | 2017MPCWPY | Luca G Guidotti |

The funders had no role in study design, data collection and interpretation, or the decision to submit the work for publication.

### Author contributions

Ngoc Lan Tran, Conceptualization, Validation, Investigation, Visualization, Methodology, Writing – original draft; Lorena Maria Ferreira, Blanca Alvarez-Moya, Conceptualization, Validation, Investigation, Visualization, Methodology, Writing – original draft, Writing – review and editing; Valentina Buttiglione, Investigation, Visualization, Methodology; Barbara Ferrini, Paola Zordan, Andrea Monestiroli, Claudio Fagioli, Investigation, Methodology; Eugenia Bezzecchi, Giulia Maria Scotti, Data curation, Methodology; Antonio Esposito, Resources, Supervision; Riccardo Leone, Chiara Gnasso, Methodology; Andrea Brendolan, Resources, Methodology; Luca G Guidotti, Resources, Funding acquisition, Writing – review and editing; Giovanni Sitia, Conceptualization, Resources, Formal analysis, Supervision, Funding acquisition, Validation, Investigation, Visualization, Methodology, Writing – original draft, Project administration, Writing – review and editing

Author ORCIDs
Ngoc Lan Tran http://orcid.org/0000-0003-4270-376X
Lorena Maria Ferreira http://orcid.org/0000-0002-4262-0406
Blanca Alvarez-Moya http://orcid.org/0000-0003-2573-7909
Paola Zordan http://orcid.org/0000-0001-6915-5535
Andrea Monestiroli http://orcid.org/0000-0001-5818-2028
Claudio Fagioli http://orcid.org/0000-0001-9889-4695
Eugenia Bezzecchi http://orcid.org/0000-0002-8838-4736
Chiara Gnasso http://orcid.org/0000-0001-8005-5439
Andrea Brendolan http://orcid.org/0000-0003-0109-6879
Luca G Guidotti http://orcid.org/0000-0002-0205-2678
Giovanni Sitia http://orcid.org/0000-0003-1024-9128

## Ethics

All animal experiments were approved by the Animal Care and Use Committee of the San Raffaele Scientific Institute (IACUC 691, 808 and 1042) and were conducted in specific pathogen-free (SPF) facility in microisolator cages under a 12-hour light/dark cycle with free access to water and standard mouse diet (Teklad Global 18% Protein Rodent Diet, Harlan).

## Decision letter and Author response

Decision letter https://doi.org/10.7554/eLife.80690.sa1
Author response https://doi.org/10.7554/eLife.80690.sa2

# Additional files

## Supplementary files
• Transparent reporting form

## Data availability

Sequencing data have been deposited in the Gene Expression Omnibus (GEO) database under the accession number GSE186203 for intrahepatic CD31 bulk RNA-seq. All data generated or analyzed during this study are included in the manuscript and supporting files.

The following dataset was generated:

| Author(s) | Year | Dataset title | Dataset URL | Database and Identifier |
|---|---|---|---|---|
| Ferreira LM, Alvarez-Moya BA, Bezzecchi E, Scotti GM, Sitia G | 2021 | Continuous sensing of IFNα by hepatic endothelial cells shapes a vascular antimetastatic barrier | https://www.ncbi.nlm.nih.gov/geo/query/acc.cgi?acc=GSE186203 | NCBI Gene Expression Omnibus, GSE186203 |

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

# Appendix 1

## Appendix 1—key resources table

| Reagent type (species) or resource | Designation | Source or reference | Identifiers | Additional information |
|---|---|---|---|---|
| Strain, strain background (*Mus musculus*) | C57BL/6 J | Charles River, Italy | Strain code: 632 | Inbred |
| Strain, strain background (*Mus musculus*) | BALB/c | Charles River Lab | Strain code: 028 | Inbred |
| Strain, strain background (*Mus musculus*) | NSG (NOD.Cg-Prkdcscid Il2rgtm1Wjl/SzJ) | Charles River Lab | Strain code: 614 | Immun odeficient |
| Strain, strain background (*Mus musculus*) | *Ifnar1*$^{fl/fl}$ (B6(Cg)-*Ifnar1*tm1.1Ees/J) | Jackson Lab | *JAX:028256* | C57BL/6 J background |
| Strain, strain background (*Mus musculus*) | *Alb*$^{Cre}$ (B6.Cg-Speer6-ps1Tg(*Alb*-cre)21Mgn/J) | Jackson Lab | JAX:003574 | C57BL/6 J background |
| Strain, strain background (*Mus musculus*) | *Pdgfrb*$^{CreERT2}$ (B6.Cg-Tg(Pdgfrb-cre/ERT2)6096Rha/J) | Jackson Lab | JAX:029684 | C57BL/6 J background |
| Strain, strain background (*Mus musculus*) | *Itgax*$^{Cre}$ (CD11c, B6.Cg-Tg(*Itgax*-cre)1-1Reiz/J) | Jackson Lab | JAX:008068 | C57BL/6 J background |
| Strain, strain background (*Mus musculus*) | Rosa26-ZsGreen (B6.Cg-Gt(ROSA)26Sortm6(CAG-ZsGreen1)Hze/J) | Jackson Lab | JAX:007906 | C57BL/6 J background |
| Strain, strain background (*Mus musculus*) | *VeCad*$^{CreERT2}$ (*Cdh5*(PAC)$^-$$^{CreERT2}$ mice) | Taconic, Models for life | Provided by S. Brunelli (UniMib, Milan) | C57BL/6 J background |
| Strain, strain background (*Mus musculus*) | OT-I mice (C57BL/6-Tg(TcraTcrb)1100Mgb/Crl) | Charles River Lab | JAX:003831 | C57BL/6 J background |
| Antibody | Anti-CD126, IL-6RA, (clone: D7715A7) (Rat monoclonal) | Biolegend | Cat# 115806, RRID:AB_313676 | FC (1:100) |
| Antibody | anti-CD18, ITGB2 (clone M18/2) (Rat monoclonal) | Biolegend | Cat# 101402, RRID:AB_312811 | FC (1:100) |
| Antibody | Anti-CD49d, ITGA4, (clone 9C10(MFR4.B)) (Rat monoclonal) | Biolegend | Cat# 103706, RRID:AB_313046 | FC (1:100) |
| Antibody | Anti-H-2K$^b$/H-2D$^b$ (clone 28-8-6) (Mouse monoclonal) | Biolegend | Cat# 114605, RRID:AB_313596 | FC (1:100) |
| Antibody | Anti-CD86 (clone GL-1) (Rat monoclonal) | Biolegend | Cat# 105007, RRID:AB_313150 | FC (1:100) |
| Antibody | Anti-CD146 (clone ME-9F1) (Rat monoclonal) | Biolegend | Cat# 134704, RRID:AB_2143527 | FC (1:100) |
| Antibody | Anti-CD44 (clone IM7) (Rat monoclonal) | Biolegend | Cat# 103044, RRID:AB_2650923 | FC (1:100) |
| Antibody | Anti-CD62L (clone MEL-14) (Rat monoclonal) | Biolegend | Cat# 104420, RRID:AB_493376 | FC (1:100) |
| Antibody | Anti- CD16/CD32, (Rat monoclonal) | BioXCell | Cat# BE0307; RRID:AB_2736987 | FC (1:100) |
| Antibody | Anti-GFP, (Rabbit polyclonal) | Thermo Fisher Sci | Cat# A11122, RRID:AB_221569 | IF (1:100) |
| Antibody | Anti-GFP (clone FM264G) (Rat monoclonal) | Biolegend | Cat# 338008, RRID:AB_2563288 | IF (1:100) |
| Antibody | Anti-CD31, PECAM1, (Goat polyclonal) | R&D Systems | Cat# AF3628, RRID:AB_2161028 | IF (1:300) |

*Appendix 1 Continued on next page*

*Appendix 1 Continued*

| Reagent type (species) or resource | Designation | Source or reference | Identifiers | Additional information |
|---|---|---|---|---|
| Antibody | Anti-PDGFRβ, CD140b (clone APB5) (Rat monoclonal) | eBioscience | Cat# 14-1402-82, RRID:AB_467493 | IF (1:200) |
| Antibody | Anti-CD11c, (clone N418) (Armenian hamster monoclonal) | Biolegend | Cat#117312, RRID:AB_492850 | IF (1:100) |
| Antibody | Anti-Heparan Sulfate (clone F58-10E4) (Mouse monoclonal) | Amsbio | Cat# 370255 S, RRID:AB_10891554 | IF (1:50) |
| Antibody | Anti-LYVE-1 (Rabbit polyclonal) | Novus Biol | Cat# NB600-1008, RRID:AB_10000497 | IF (1:300) |
| Antibody | Anti-Collagen type IV (Rabbit polyclonal) | Abcam | Cat# ab19808, RRID:AB_445160 | IF (1:100) |
| Antibody | Anti-Laminin (Rabbit polyclonal) | Sigma-Aldrich | Cat# L9393, RRID:AB_477163 | IF (1:300) |
| Antibody | Anti-CD54, ICAM1 (clone YN1/1.7.4) (Rat monoclonal) | Biolegend | Cat#116101, RRID:AB_313692 | IF (1:100) |
| Antibody | Anti-CD62E, E-selectin (clone 10E9.6) (Rat monoclonal) | BD Bioscience | Cat#550290, RRID:AB_393585 | IF (1:100) |
| Antibody | Anti-Rabbit AF488 (Donkey polyclonal) | Thermo Fisher Sci | Cat#32790, RRID:AB_2762833 | IF (1:200) |
| Antibody | Anti-Goat AF546 (Donkey polyclonal) | Thermo Fisher Sci | Cat#A-11056, RRID:AB_2534103 | IF (1:200) |
| Antibody | Anti-Rat AF647 (Donkey polyclonal) | Jackson IR | Cat#712-605-153, RRID:AB_2340694 | IF (1:200) |
| Antibody | Anti-IgM conjugated to APC (clone II/41) (Rat monoclonal) | Thermo Fisher Sci | Cat#17-5790-82, RRID:AB_469458 | IF (1:100) |
| Antibody | Anti-Ki-67 (clone SP6) (Rabbit recombinant antibody) | Thermo Fisher Sci | Cat# MA5-14520, RID:AB_10979488 | IHC (1:200) |
| Antibody | Anti-CD34 (clone MEC14.7) (Rat monoclonal) | Biolegend | Cat# 119301, RRID:AB_345279 | IHC (1:300) |
| Antibody | Anti-F4/80 (clone A3-1) (Rat monoclonal) | Bio-Rad | Cat# MCA497, RRID:AB_2098196 | IHC (1:200) |
| Antibody | Anti-CD3 (clone SP7) (Rabbit monoclonal) | Abcam | Cat# ab21703, RRID:AB_446487 | IHC (1:100) |
| Antibody | Anti-pSTAT1 (Clone 58D6) (Rabbit monoclonal) | Cell Signaling | Cat# BK9167S, RRID:AB_561284 | IHC (1:800) |
| Antibody | Anti-CD11c microbeads ultrapure | Miltenyi Biotec | Cat# 130-125-835 | 100 µL per $10^8$ total cells |
| Antibody | Anti-F4/80 microbeads ultrapure | Miltenyi Biotec | Cat# 130-110-443 | 100 µL per $10^8$ total cells |
| Antibody | InVivoMAb Anti-mouse GM-CSF(clone MP1-22E9) (Rat monoclonal) | BioXcell | Cat# BE0259 RRID:AB_2687738 | 40 ng/ml |
| Peptide, recombinant protein | Recombinant mouse IFNα1, carrier-free | Biolegend | Cat# 751802 | |
| Peptide, recombinant protein | NH2-SIINFEKL-COOH (SL-8 peptide) | Proimmune | Cat# GMPT5995 | |
| Chemical compound, drug | Albumin from chicken egg white (ovalbumin) | Sigma-Aldrich | Cat# 55039 | |
| Chemical compound, drug | Gadoxetic acid (Primovist; Gd-EOB-DTPA) | Bayer Schering Pharma | N/A | |

*Appendix 1 Continued on next page*

*Appendix 1 Continued*

| Reagent type (species) or resource | Designation | Source or reference | Identifiers | Additional information |
|---|---|---|---|---|
| Chemical compound, drug | Tamoxifen | Sigma Aldrich | Cat# T5648-5G | |
| Chemical compound, drug | Lipofectamine 2000 | Thermo Fisher Sci | Cat# 11668019 | |
| Chemical compound, drug | Paraformaldehyde powder | Sigma-Aldrich | Cat# 158127 | |
| Chemical compound, drug | Sodium Cacodylate powder | Sigma-Aldrich | Cat# C0250 | |
| Chemical compound, drug | 25% EM grade Glutaraldehyde | ProSciTech | Cat# C001 | |
| Chemical compound, drug | Osmium tetroxide | ProSciTech | Cat# C011 | |
| Chemical compound, drug | Lanthanum (III) nitrate hexahydrate | MERK | Cat# 331937–11 G | 2% |
| Chemical compound, drug | Zinc formalin fixative | Sigma-Aldrich | Cat# Z2902-3.75L | |
| Chemical compound, drug | Fibronectin human plasma | MERK | Cat# F2006-1MG | |
| Chemical compound, drug | Percoll | Sigma-Aldrich | Cat# P4937-500ML | |
| Chemical compound, drug | Matrigel Matrix | BD Bioscience | Cat# 354248 | |
| Chemical compound | 7-AAD Viability staining solution | Biolegend | Cat# 420404 | FC (1:100) |
| Chemical compound, drug | EGM-2MV endothelial cell growth medium-2 BulletKit | Lonza | Cat# CC-3202 | |
| Chemical compound, drug | Brefeldin A | Sigma-Aldrich | Cat# B7651 | |
| Commercial assay, kit | VeriKine-HS Mouse IFNα All-Subtype ELISA Kit | PBL Assay Sci | Cat# 42115–1 | |
| Commercial assay, kit | GeneArt CRISPR Nuclease Vector with OFP Reporter Kit | Thermo Fisher Sci | Cat# A21174 | |
| Commercial assay, kit | TOPO TA Cloning Kit | Thermo Fisher Sci | Cat# 450641 | |
| Commercial assay, kit | CellTiter-Glo Luminescent Cell Viability Assay | Promega | Cat# G7570 | |
| Commercial assay, kit | ReliaPrep RNA Cell Miniprep System | Promega | Cat# Z6011 | |
| Commercial assay, kit | ReliaPrep RNA Tissue Miniprep System | Promega | Cat# Z6111 | |
| Commercial assay, kit | TURBO DNA-free kit | Ambion | Cat# AM1907 | |
| Commercial assay, kit | QIAamp DNA Mini kit | Qiagen | Cat# 51304 | |
| Commercial assay, kit | RNeasy Mini Kit | Qiagen | Cat# 74104 | |
| Commercial assay, kit | Procyte kit | Idexx | Cat# 9926306–00 | |
| Commercial assay, kit | Live/Dead fixable near-IR dead cell stain kit | Thermo Fisher Sci | Cat# L34975 | FC (1:500) |
| Commercial assay, kit | Live/Dead fixable Green dead cell stain kit | Thermo Fisher Sci | Cat# L34970 | FC (1:100) |
| Commercial assay, kit | DNAse TURBO | Thermo Fisher Sci | Cat# AM1907 | |

*Appendix 1 Continued on next page*

*Appendix 1 Continued*

| Reagent type (species) or resource | Designation | Source or reference | Identifiers | Additional information |
|---|---|---|---|---|
| Commercial assay, kit | Foxp3/Transcription Factor Staining buffer set | Thermo Fisher Sci | Cat# 00-5523-00 | |
| Cell lines (*Mus musculus*) | CT26 | ATCC | Cat# CRL-2638, RID:CVCL_7256 | |
| Cell lines (*Mus musculus*) | MC38 | Kindly provided by P. Berraondo López - Centro de Investigación Médica Aplicada (Spain) | RRID:CVCL_B288 | |
| Sequence-based reagent | *Ifnar1* intron 3 forward | *Prigge et al., 2015* | PCR primers | ACTCAGGTTCG CTCCATCAG |
| Sequence-based reagent | *Ifnar1* intron 3 reverse | *Prigge et al., 2015* | PCR primers | GCACATTGACCATT ACAAGAGTAG |
| Sequence-based reagent | *Ifnar1* exon2 forward | This paper | PCR primers | TCCAAGACTCC TGCTGTC |
| Sequence-based reagent | *Ifnar1* exon2 reverse | This paper | PCR primers | GCACTTTTAC TTGCTCGGT |
| Sequence-based reagent | *Itgax*Cre forward | Jackson Lab | PCR primers | ACTTGGCAGCTG TCTCCAAG |
| Sequence-based reagent | *Itgax*Cre reverse | Jackson Lab | PCR primers | GTGGCAGATGG CGCGGCA |
| Sequence-based reagent | *ALB*Cre forward | Jackson Lab | PCR primers | CCAGGCTAAGTGCC TTCTCTACA |
| Sequence-based reagent | *ALB*Cre reverse | Jackson Lab | PCR primers | AATGCTTCTGTCC GTTTGCCGGT |
| Sequence-based reagent | *Pdgfrb*Cre forward | Jackson Lab | PCR primers | GAACTGTCACC GGGAGGA |
| Sequence-based reagent | *Pdgfrb*Cre reverse | Jackson Lab | PCR primers | AGGCAAATTTT GGTGTACGG |
| Sequence-based reagent | *VeCad*Cre forward | *Wang et al., 2010* | PCR primers | GCCTGCATTACCG GTCGATGCAACG |
| Sequence-based reagent | *VeCad*Cre reverse | *Wang et al., 2010* | PCR primers | GTGGCAGATGGCG CGGCAACACCAT |
| Sequence-based reagent | RosaZsgreen forward | Jackson Lab | PCR primers | AACCAGAAGTGG CACCTGAC |
| Sequence-based reagent | RosaZsgreen reverse | Jackson Lab | PCR primers | GGCATTAAAGC AGCGTATCC |
| Sequence-based reagent | crRNA1 | Thermo Fisher Sci | CRISPR RNA guide | TAGACGTCTATATT CTCAGGGTTTT |
| Sequence-based reagent | crRNA2 | Thermo Fisher Sci | CRISPR RNA guide | ATGTAGACGTCTAT ATTCTCGTTTT |
| Sequence-based reagent | Mm00516793_g1 | Thermo Fisher Sci | Taqman probe *Irf7* | |
| Sequence-based reagent | Mm00836412_m1 | Thermo Fisher Sci | Taqman probe *Oas1* | |
| Sequence-based reagent | Mm 99999915_g1 | Thermo Fisher Sci | Taqman probe *Gapdh* | |
| Software and algorithms | FlowJo v10.5 or greater | Tree Star | RRID:SCR_008520 | |
| Software and algorithms | Prism version 8 or greater | GraphPad | RRID:SCR_002798 | |
| Software and algorithms | ImageScope software | Leica Biosystems | RRID:SCR_014311 | |
| Software and algorithms | Imaris version 7.2.3 | Bitplane | RRID:SCR_007370; | |
| Software and algorithms | Fiji software or ImageJ software | NIH | RRID:SCR_002285; 003070 | |

*Appendix 1 Continued on next page*

*Appendix 1 Continued*

| Reagent type (species) or resource | Designation | Source or reference | Identifiers | Additional information |
|---|---|---|---|---|
| Software and algorithms | MIPAV v5.3.4 or greater | CIT, NIH | RRID:SCR_007371 | |
| Software and algorithms | EsivisionPro 3.2 software | Soft Imaging Sys | N/A | |
| Software and algorithms | OsiriX DICOM viewer version 3.9.2 or greater | Pixmeo SARL | N/A | |
| Software and algorithms | Trimmomatic version 0.32 | *Bolger et al., 2014* | RRID:SCR_011848 | |
| Software and algorithms | STAR aligner version 2.5.3 a | STAR SRL | RRID:SCR_004463 | |
| Software and algorithms | GENECODE version M22 | Ensmbl 97 | N/A | |
| Software and algorithms | R-4.0.3 software | R Core Team | N/A | |
| Software and algorithms | DESeq2 version 1.30.1 | *Love et al., 2014* | N/A | |
| Software and algorithms | EnrichR R package version 3.3 | *Chen et al., 2013* | RRID:SCR_020938 | |
| Software and algorithms | GeneArt CRISPR Search and Design Tool | Thermo Fisher Sci | N/A | |
| Other | Alzet osmotic pumps 1002 | Alzet | Cat# 10104844 | Mini osmotic pump (MOP) |
| Other | Alzet osmotic pumps 1004 | Alzet | Cat# 10104846 | Mini osmotic pump (MOP) |
| Other | ProCyte Dx Hematology Analyzer | IDEXX Lab | N/A | Instruments |
| Other | Bruker Horizontal 7-Tesla MRI scanner | BioSpec | N/A | Instruments |
| Other | ViiA7 Fast Real-Time PCR System | Applied Biosys | Cat# 4453543 | Instruments |
| Other | Aperio Scanscope XT Leica | Leica Biosystems | RRID:SCR_018457 | Instruments |
| Other | Leica SP8 LIGHTNING confocal microscope | Leica Biosystems | RRID:SCR_018169 | Instruments |
| Other | FEI Talos L120C G2 Transmission electron microscope | Thermo Fisher Sci | N/A | Instruments |
| Other | Zeiss FEG Gemini 500 Scanning electron microscope | Zeiss | N/A | Instruments |
| Other | FACSCanto II High Throughput Sampler | BD Bioscience | N/A | Instruments |
| Other | FACS LSRFortessa | BD Bioscience | N/A | Instruments |
| Other | FACSAria Fusion | BD Bioscience | N/A | Instruments |
| Other | Illumina NovaSeq 6000 machine | Illumina | RRID:SCR_016387 | Instruments |
| Other | Leica CM1520 Cryostat | Leica Biosystems | RRID:SCR_017543 | Instruments |
| Other | Leica VT1000S Vibratome | Leica Biosystems | RRID:SCR_016495 | Instruments |
| Other | EM UC7 ultramicrotome | Leica Biosystems | RRID:SCR_016694 | Instruments |

