## [Editor Report]

This study describing how continuous perioperative IFNα therapy stimulates hepatic endothelial cells to build up a physical vascular barrier that limits tumor cell entry into the liver and promotes long-term antitumor immunity provides novel evidence for anti-metastatic effects of IFNα via effects on the liver vascular compartment. This work is predicted to advance the field and will be of interest to scientists studying cancer, inflammation and liver function.

---

## [Decision Letter]

[Editors' note: this paper was reviewed by Review Commons.]

---

## [Author Response]

We thank the referees for their reviews and helpful comments. We have revised our manuscript based on these comments, adding new experimental data that should address the referees’ concerns.

Response to Reviewer #1:1. Authors use an elegant orthotopic model of liver metastasis to confirm the effect of continuous IFNα on hepatic colonization (Figure 3). Although they extensively characterize the metastatic lesions, they do not show data on the potential impact of IFNα treatment in the primary caecum tumour. Authors should clarify if the described effects are taken place in the liver or/and in the caecum. It would be interesting to show if IFNα affects the primary tumour size, the extravasation of cancer cells and the immune infiltration since all these factors could have an impact in the number of liver lesions.

We thank the reviewer for acknowledging the importance of our results particularly in the context of the orthotopic mouse model we developed. We agree that displaying the results of continuous IFNα therapy on primary intracecal tumors, as well as the results pertaining to the few mice that develop microscopic or macroscopic liver metastasis, is important for the interpretation of our work. Thus, we evaluated the dimension of primary intracecal CRC lesions (Figure 3D,E) and we performed additional IHC characterization of the primary tumors (Figure S4A,B). The analysis showed that the dimension of the primary lesions and the markers we analyzed were non significantly modified by continuous IFNα therapy (Figure 3D,E and Figure S4A,B). These results favor the hypothesis that IFNα therapy does not modify the number of cells that spread from the primary tumors and seed into the liver, but it rather impinges on the intravascular containment of CRC cells circulating within the liver (Figure 3F). As said earlier, the data also highlight the possibility that CRC tumors may become refractory to IFNα or that the dose and schedule we adopted does not significantly affect the growth of established liver CRCs at late time points. The data are also consistent with results obtained with MC38^*Ifnar1*_KO^ CRC cells indicating that continuous IFNα therapy does not require *Ifnar1* expression by tumor cells to exert its antimetastatic function (Figure 4A,C-D). This is also in line with the high IFNα concentrations required to activate the "tunable" direct antiproliferative functions of this cytokine that exceed those achieved in our system (Catarinella *et al.,* 2016; Schreiber, 2017).

Text has been added in the revised manuscript at lines 175-197 and in the discussion lines 425-431.

2. Figure 3f right shows liver images without any obvious metastatic lesion. Since authors are analysing the effect of IFNα treatment in proliferation, vascularization and immune composition in liver tumours, they may show and quantify images with metastatic lesions and restrict the analysis to the tumour area.

Since the main finding of our manuscript regards the prevention of hepatic colonization by continuous IFNα therapy, we think that the original data presented in Figure 3G,H are representative of the overall efficacy of our strategy that confers protection in up to 60% of the mice carrying intramesenteric tumors of increasing dimensions (Figure 3H). We have thus maintained our original results, adding the quantification of all IHC data on groups of Sham control livers (n=6), as suggested. In any case, we also included the same IHC characterization of the few and small intrahepatic lesions that have bypassed the intravascular antimetastatic barrier (Figure S4C,D). Indeed, in agreement with the results observed in primary intracecal lesions, these metastatic lesions that developed in IFNαtreated mice showed similar markers of cell proliferation, neoangiogenesis, F4/80 macrophages and CD3^+^ T cells, as control lesions detected in NaCl^-^treated mice. Once again, the results highlight the possibility that CRC tumors, once established as micro/macroscopic metastases, may become refractory and resistant to IFNα therapy by downregulating the *Ifnar1* in various components of the tumor microenvironment (Boukhaled *et al.*, 2021; Katlinski *et al.*, 2017). Text has been added in the revised manuscript at lines 175-197 and in the discussion lines 496-515.

3. Authors analyse the recombination efficiency of different mouse CRE lines by nonquantitative methods (PCR of hepatic genomic DNA and GFP expression by immunofluorescence in healthy liver). Since PDGFRβ-Cre/ERT2 and CD11c-Cre lines are used to exclude a role of IFNα on the targeted cells, authors should provide stronger evidences to support this. They may consider studing the ablation of Ifnar1 in FACS sorted fibroblasts and myeloid cells. Moreover, it would be important showing the proportion of GFP+ cells in the sorted populations to understand how broadly these stromal populations are targeted.

We thank the referee for raising this important issue, which is related to the relative efficiency of *Ifnar1* recombination in each of the Cre-expressing mouse models we have used in the study. To this regard, we newly performed an extensive colocalization analysis quantifying the percentage of GFP^+^ cells that colocalize with cell specific markers (i.e., PDGFRβ, CD11c, F4/80 and CD31) of the various mouse models (PDGFRβ^CreERT2^, CD11c^Cre^ and VeCad^CreERT2^, respectively) crossed with RosaZsGreen reporter mice.

Colocalization analysis of GFP in the different systems was performed using the ImageJ “colocalization” algorithm developed by Pierre Bourdoncle (Institut Jacques Monod, Service Imagerie, Paris; 2003–2004). The method allows the generation of unsupervised profiles of co-localized pixels between two channels. This methodology has been included in the section Methods and Protocols, line 806-809. Of note, we observed an almost complete recombination in liver fibroblast (GFP^+^/PDGFRβ^+^), with about 98.2 ± 0.72% hepatic stellate cells that co-expressed GFP^+^ and PDGFRβ^+^ signals (see the new Figure S5E). Similarly, hepatic DCs (GFP^+^/CD11c^+^) had 94.17 ± 2.16% colocalization, while F4/80^+^ KCs or LCMs (GFP^+^/F4/80^+^) colocalized in 78.14 ± 5.03% (see the new Figure S5E). Finally, HECs, including LSECs, (GFP^+^/CD31^+^) showed 85.3 ± 5.03% colocalization (see the new Figure S5E,F), with no expression of GFP signals in cells other than CD31^+^. Note that these values indicate an almost complete colocalization of the Cre recombinase in the target cell types analyzed (see representative IF shown in Figure S5E). Text has been added in the revised manuscript at lines 225-233.

Moreover, DEGs analysis between NaCl^-^treated VeCad*^Ifnar1_KO^* and *Ifnar1^fl/fl^* HECs showed a significant downregulation of *Ifnar1* expression in CD31+ VeCad*^Ifnar1_KO^* cells, with a log_2_ fold-change of -0.387 and an adjusted p-value of 0.033, further confirming Cre recombination in HECs isolated from VeCad*^Ifnar1_KO^* mice (as depicted in the heatmap of Figure 6B; the 12^th^ gene of the Type I IFN response is *Ifnar1*).

We have prepared all source images at higher dimension to better appreciate the colocalization within liver microvasculature. In addition, we performed several flow cytometry analyses to identify liver cell populations of Cre-recombinant mice that express *Ifnar1*. Unfortunately, the predicted low cellular surface expression of this molecule coupled with the experimental conditions needed to extract viable non-parenchymal cells from the liver have prevented us from obtaining informative results*.*

4. Ifnar1 ablation in VeCad+ cells prevents the effect of IFNα on tumour growth (Figure 4d), suggesting the existence of anti-tumour mechanisms beyond the effects on hepatic colonization. Authors may consider checking proliferation, vascularization and immune infiltration in these tumours to enhance their conclusion.

We fully agree with the referee’s concern and as above mentioned, we have followed his/her suggestion and examined the existence of antitumor mechanisms beyond the effects on hepatic colonization in VeCad^*Ifnar1*_KO^ mice treated with NaCl or IFNα. To this end, 4 NaCl^-^Ifnar1^fl/fl^, 7 IFNα-Ifnar1^fl/fl^, 4 NaCl^-^VeCad^Ifnar1_KO^ and 4 IFNα-VeCad^Ifnar1_KO^ mice were intrasplenically injected with MC38 CRC cells (Figure S7A,B). Twenty-one days after injection, mice were euthanized and their livers analyzed for tumor size, proliferation, signs of angiogenesis (as denoted by CD34 staining) and immune infiltration (F4/80^+^ macrophages and CD3^+^ T cells). Consistent with data presented in Figure 4D, histological analysis showed that *Ifnar1*^fl/fl^ mice did not develop liver metastases in IFNα-treated mice. Furthermore, metastatic lesions detected in VeCad^*Ifnar1*_KO^ mice treated or not with IFNα did not show significant differences in Ki67 positivity, CD34 staining or the amount of F4/80^+^ resident macrophages and CD3^+^ T cells. This further supports that the antimetastatic potential of IFNα therapy may be primarily depend on the inhibition of hepatic trans-sinusoidal migration, a limiting step in the metastatic cascade that could secondarily influence colonization and outgrowth (Chambers *et al.,* 2002). Corresponding text has been added at lines 248-252.

5. Immune properties of LSECs are analysed in vivo by using a mouse CRE line that targets all endothelial cells, including those ones located in lymphoid organs, and evaluating T cell composition in the spleen. I found difficult to conclude that these properties are exerted directly by LSECs and not by other endothelial cells in vivo. To clarify the local effect of LSECs in modulating anti-tumour immunity, T cell composition and activation should be checked in tumours shortly after tamoxifen administration.

We thank the reviewer for pointing out this issue, which cannot not be tested directly because – as also mentioned by reviewer 2 – LSEC-specific Cre-recombinant driver mice do not exist. As also indicated in the cited literature, central memory T cells accumulate after peripheral priming in secondary lymphoid organs such as the spleen (Sallusto *et al.,* 2004; Stone *et al.,* 2009; Yu *et al.,* 2019). To this end, the generation and regulation of antitumor immunity is a highly orchestrated multistep process involving the uptake of tumor-associated antigens by professional APCs, their time-consuming migration to draining lymph nodes and the generation of protective T cells. Unlike other APCs, HECs/LSECs do not need to migrate to draining lymph nodes to activate effector T cells, leading to a rapid intrahepatic CD8^+^ T cell activation.

In this context, LSECs must not only efficiently uptake, process and present CRC-derived antigens coming from intravascularly contained tumor cells, but they also require the attraction and retention within the liver micro-vasculature of T cell populations necessary for the generation of effective antitumor immune responses, where chemokines play an important role (Lalor *et al.,* 2002). As shown in Figure 6A-C, two prominent chemokines (*Cxcl10* and *Cxcl9*) required for T cell recruitment to the liver are specifically upregulated only in HECs/LSECs from IFNα-treated *Ifnar1*^fl/fl^ mice, whereas HECs from VeCad^Ifnar1_KO^ mice maintained low expression of these chemoattractants in both NaCl^-^ and IFNα-treated mice.

These data are also consistent with the in vitro cross-priming results (see Figure 7A,B) showing that in the absence of IFNα, HECs have a low capacity to prime naïve T cells (Katz *et al.,* 2004), indicating that LSEC-primed by tumor-derived antigens coming from apoptotic intravascular CRC metastatic cells play an important role in inducing tolerance (Berg *et al.,* 2006; Katz *et al.*, 2004), especially when CRC cells quickly extravasate and position within the space of Disse, likely becoming less accessible to intravascular patrolling by naïve and effector T cells (Benechet *et al.,* 2019; Guidotti *et al.,* 2015). On the contrary, in IFNα-treated Ifnar1^fl/fl^ mice, CRC cells are rapidly contained in the liver microvasculature (Figure 5A,B) with CRC-derived antigens that could be immediately taken up by LSECs due to their anatomical proximity and efficient endocytosis capacity, which is among the highest of all cell types in the body (Sorensen, 2020). Here, the continuous sensing of IFNα by LSECs upregulates several genes related to antigen processing and presentation pathways (Figure 6B,D), leading to efficient cross-priming of tumor-specific CD8^+^ T cells to the same extent as professional APCs, such as splenic DCs (Figure 7B).

Text has been added in the revised manuscript at lines 496-515.

Finally, regarding the suggestion to analyze the role of HECs/LSECs in inducing antitumor T cell immunity shortly after tamoxifen administration, while we agree that it would be interesting to analyze HEC/LSEC-mediated T cell activation by treating NaCl^-^ and IFNαtreated *Ifnar1*^fl/fl^ and VeCad^Ifnar1_KO^ mice with tamoxifen after CRC cell injection, we would like to point out that tamoxifen treatment will not only induce Cre recombination and *Ifnar1* loss on endothelial cells but it may also induce several “off-target” effects complicating the interpretation of the results. Indeed, tamoxifen is known to (i) inhibit the in vitro proliferation of several CRC cell lines (Ziv *et al.,* 1994), (ii) impair the growth of CRC liver metastases in vivo (Kuruppu *et al.,* 1998) and (iii) modify matrix stiffness to reduce tumor cell survival (Cortes *et al.,* 2019). Further, as IFNα modifies the hepatic vascular barrier and the accessibility of antigens by LSECs, the specific timing of tamoxifen treatment could also affect the immunological consequences of *Ifnar1* deletion making these experiment impractical. For these reasons, we’d like not to perform the suggested experiment with tamoxifen.

Response to Reviewer #21. First, the authors started their experiments with MC38 and CT26 CRC cell lines. At the end they just applied MC38. The rational behind this should be clearly stated. Second, as in their previous publication (Catarinella et al., 2016) F1 hybrids of C57BL/6 x BALB/c mice were used for the experiments. However, I believe that the genetic heterogeneity might be strongly increased by this approach which might lead to difficult reproducibility of the results.

We thank the referee for raising this important issue; additional text describing the reason of our choice has been introduced at lines: 203-205. We respectfully disagree with the comment that CB6F1 hybrids may increase genetic heterogeneity and impair reproducibility of our results. Each CB6F1 hybrid individual is genetically identical to its littermates, sharing 50% of genes of each parental mouse line and being tolerant to reciprocal MHC-I genes (thus permitting the correct engraftment of both cell lines). We agree that the use of mismatched backcrosses after the F1 generation would increase genetic heterogeneity and thus may affect outcome. This is also the reason why we could not perform experiments with CT26 in the *Ifnar1*^fl/fl^ conditional lines that are in C57BL/6 background and would have needed at least 10 generations of backcrossing in the BALB/c background before being suitable to such experiments. Finally, all experiments described in Figure 4, 5, 6 and 7 were performed in C57BL/6 mice using MC38 CRC cells with results that reproduced those obtained in CB6F1 hybrids, and very similarly to what we have previously reported with MC38 in C57BL/6 mice (see Figure 5 (Catarinella *et al.*, 2016)).

2. At page 16 the authors conclude that "patients suffering from chronic liver fibrotic disease… display lower incidence of hepatic metastases". In the community there is contradictory data (see Kondo et al., BJC, 2016, https://www.nature.com/articles/bjc2016155). This should be precisely discussed, otherwise this claim should be removed.

We thank the referee for raising this issue and modified the discussion accordingly. Text has been added in the revised manuscript at lines 455-457.

3. In the Discussion section the interplay of other cell types within the hepatic niche should be stated. For example, in Toyoshima's study a direct anti-tumoral effect of dendritic cells releasing IFNα1 was demonstrated (see Toyoshima et al., Cancer Immunol Res, 2019, https://aacrjournals.org/cancerimmunolres/article/7/12/1944/469540/IL6-Modulatesthe-Immune-Status-of-the-Tumor). This further strengthens your data.

We agree with the reviewer's suggestion and added new text to recognized the interplay between different cell types such as dendritic cells within the hepatic niche (see new text at lines 505-515).

4. Last, multiple times the authors write about data that is "not shown". Please either include these data in the manuscript or delete corresponding phrases because it is not possible for the reader to scrutinize it.

We fully agree with the referee’s concern and displayed all “not shown results” in Figure S1E and Figure S9C-I.

5. Besides, I suggest additional experiments further substantiating the study: To see if this effect of IFNα1 is cell type-specific liver metastasis of other solid tumors such as breast cancer or melanoma should be investigated.

We agree with the reviewer's suggestion, as also indicated in our original discussion. We believe that additional experiments with other solid tumor cell lines would be important to generalize the potential of perioperative IFNα therapy. In particular, we believe that pancreatic ductal adenocarcinoma (PDAC), a highly lethal disease that most commonly metastasizes to the liver (Lambert *et al.,* 2017), may benefit from our approach. It should be noted, however, that the pleotropic nature of IFNα allows this cytokine to inhibit tumor growth by several mechanisms. Above all, the ability of IFNα therapy to directly reduce tumor growth depends on the relative surface expression of Ifnar1 on each tumor cell and the ability to maintain such expression in the harsh tumor microenvironment during IFNα therapy. As the degradation of Ifnar1 by CRC tumors has been well described (Katlinski *et al.*, 2017), it is possible that CRC tumors thus escaping the antitumor properties of endogenous type I interferons may respond less efficiently to therapeutic IFNα regimens such as those herein described. This notion is consistent with our data on primary orthotopic tumors (Figure 3D,E), which are no longer responsive to continuous IFNα therapy as early as 7 days after implantation of CT26^LM3^ cells. In addition, the definition of the HEC/LSEC antimetastatic barrier has been possible only because CRC cells are not directly susceptible to the IFNα antiproliferative activity, which we observed in vitro at extremely high IFNα dosages (Catarinella *et al.*, 2016) but not in vivo (as formally demonstrated by using MC38*^Ifnar_ko^* cells, Figure 4A). Properly addressing the reviewer’s comment would thus require extensive investigations involving the establishment of new mouse models of metastases from other solid tumors, starting from the in vitro and in vivo regulation of surface Ifnar1 expression in each tumor cell. We strongly believe that this work has merit but we think that it should be reported separately.

6. The authors applied a broad range of cell type-specific mice. However, a thorough characterization of the deletion of Ifnar1 in the corresponding cell types is missing. This is crucial for the manuscript.

We fully agree with the referee’s concern and as previously mentioned, we have improved the characterization of Ifnar1 deletion (see response to the same critique received from reviewer 1, comment 3).

7. The capillarization of the hepatic vascular niche is a crucial point in this story. I believe that the hepatic endothelium should be further characterized by additional vascular markers.

In response to the reviewer’s suggestion, we have included in our analysis the characterization of Lyve-1, a marker of hepatic capillarization (Pandey *et al.,* 2020; Wohlfeil *et al.,* 2019). Indeed, IFNα treatment of *Ifnar1*^fl/fl^ mice significantly increased the expression of Lyve-1, whereas IFNα treatment of VeCad^*Ifnar1*_KO^ mice showed no effect (Figure S9A,B), further corroborating our findings. Text has been added in the revised manuscript at lines 291-294. To better aid readers, we have prepared high-resolution images for each IF channel and have provided these data as source date for Figure S9A.

8. Last, the data and methods appear adequately presented and experiments seem to be reproducible. Just in Figure 4 the exact number of mice and replicates are not clearly presented. Otherwise, everything is fine.

We thank the reviewer for raising this issue, which apparently was not properly described in our original submission. We have now included the exact number of mice in each experimental group in the figure legend to Figure 4.

Minor commentsOverall the text and figures are accurately presented. However, I would like to add further minor comments:9. In Figure 1 you present the IFNα dosing regimen. How do you explain the decrease in serum IFNα after day 2? Besides, the data points at day 0 should be excluded since measuring started from day 2! Why did you decide to treat for seven days until the start of the experiment? One could think 2 days might already be enough.

We thank the reviewer for raising these important points. Regarding the pharmacokineticpharmacodynamic (PK-PD) behavior of our approach, we do not believe that MOP reduced its pumping efficacy after day 2 (Theeuwes & Yum, 1976), nor that counterregulatory mechanisms, such as the induction of anti-IFNα blocking antibodies, occurred in such a short time frame (Wang *et al.,* 2001). It is neither feasible that IFNα treatment significantly downregulated *Ifnar1* in the liver (as demonstrated by pSTAT1 activation after MOP treatment in Figure S1E). Rather, our results reflect the PK-PD behavior of other long-lasting formulations of IFNα, which depend on intrinsic pharmacological properties of IFNα already described in (Jeon *et al.,* 2013). Text has been added in the revised manuscript at lines 110-112.

We also corrected the figures in which we quantified serum IFNα. Indeed, blood was drawn one day before MOP implantation rather than on the same day of surgery to avoid additional blood loss, which could be a source of unnecessary stress for the animals. Therefore, we corrected the Results section and Figure S1A-C and Figure 1A,B.

The decision to start treatment 7 days rather than 2 days before seeding was made for several reasons: (i) this study follows our previous gene/cell therapy approach, in which the time interval between reconstitution of the transduced bone marrow with Tie2-IFNα and tumor challenge was at least 7-8 weeks. We therefore thought that 7 days might be a sufficient/necessary time period to induce similar phenotypes in the liver after continuous IFNα administration; (ii) 7 days is a time frame compatible with the perioperative period in humans (Horowitz *et al.,* 2015). Furthermore, the side effects that patients may experience after IFNα therapy are generally limited to the first few days after administration, allowing patients to benefit from IFNα-induced vascular antimetastatic barriers at the time of surgery without potential side effects of IFNα. Because oncologic guidelines recommend starting adjuvant chemotherapy at least 4 weeks after surgery in stage 2-3 CRC patients at risk of later developing liver metastases (Engstrand *et al.,* 2019; van Gestel *et al.,* 2014), our proposed perioperative time frame does not even conflict with these indications (Van Cutsem *et al.,* 2016).

We have included additional text in the lines 131-132 to motivate the timing of our regimens.

10. Figure 2: Did you check for metastases in other organs than the liver at the timepoint of euthanization, e.g. lungs. In the Discussion section you talk about a potential influence of IFNα1 on other organs. Therefore, I think that the mice should be thoroughly analyzed and the data presented. The manuscript will benefit from it.

We thank the reviewer for this valuable comment. Indeed, we always check for dissemination of CRC metastases on MRI analysis and necroscopy.

As stated at lines 146-147 and 158 CRC tumors seeded in the liver vasculature after colonizing the liver do not spread to other organs such as the lungs. Indeed, CRC cells intravascularly seeded in the portal circulation, are trapped at the beginning of hepatic sinusoids because their diameter is bigger than that of liver sinusoids (Figure S8A,B). These micro-anatomic peculiarities are also thought to impede the spreading of tumor cells from periportal to centrilobular areas and to the general circulation (Catarinella *et al.*, 2016; Vidal-Vanaclocha, 2008), and this is consistent with studies showing that in CRC patients undergoing surgery the majority of CRC-derived circulating tumor cells are found in the portal vein (Deneve *et al.,* 2013).

11. Overall, MRI pictures and pictures of IHC or IF are sometimes too small to see. Please provide pictures with larger magnification or enlarge the images.

We thank you for this suggestion and we have indeed increased the size of all MRI, IHC, and IF images to the maximum that will fit within the figure. In addition, we presented the images at the highest magnification available, without making digital enlargements that would significantly reduce resolution.

12. Figure 3 F, G: immune cell infiltration in the liver was analyzed. Please compare it to untreated, tumor-free wildtype liver tissue.

We appreciated the reviewer's suggestion and included the results of six Sham mice per each marker in our analysis.

The text was added on the figure legends to Figure 3H and Figure S4B,D.

13. Figure 6: the graphs are too small to be read, especially the volcano plot and the gene names of the heatmap.

We increased the font size of genes in the volcano plots and heatmap in Figure 6A,B, as suggested.

14. Figure S6: Pictures of co-immunofluorescences are presented. For the reader it is really hard to distinguish the stainings and to identify colocalized areas. Please provide pictures with one channel to better compare the marker expression.

We thank the reviewer for pointing this out and we have tried to make each panel as large as possible to fit into a two-column figure. We have also prepared high magnification images of each channel for all immunofluorescence images, which we provide as source data. We hope that this is sufficient to help readers to interpret our results without increasing the number of main or supplementary figures.

15. From page 8 onwards (section about transgenic mice) LSEC was used as kind of synonym for hepatic endothelial cells. Since there is still no LSEC-specific driver mouse, it should be stated "hepatic endothelial cells" instead.

We agree with this suggestion and thus have indicated that the results refer to HECs but include a large majority of LSECs. Indeed, LSECs make up the majority (~89%) of the total HEC population (Su *et al.,* 2021). In addition, some SEM and TEM analyses were performed only on LSECs, as well as the IF analyses. Therefore, we believe that LSECs play an important role in this process. Although not specifically suggested, we have also changed the title of our manuscript to reflect the reviewer's suggestion. Thus, we propose "Continuous sensing of IFNα by hepatic endothelial cells shapes a vascular antimetastatic barrier" as new title.

16. P. 11: there is a typo: Figure S6G,H

We corrected this typo.

17. P. 13: the authors describe Gata4 as inhibitor of subendothelial matrix deposition. This should be precisely written, since Gata4 originally is described as master-regulator of liver sinusoidal differentiation which leads to liver fibrosis development upon loss of Gata4. Besides, I came across a study of the same group that investigated the role of Notch signaling in hepatic CRC and melanoma metastasis (Wohlfeil et al., Cancer Res, 2019, https://aacrjournals.org/cancerres/article/79/3/598/638600/Hepatic-EndothelialNotch-Activation-Protects). Similar to your study they tie the reduction in hepatic metastasis to capillarization of the hepatic microvasculature.

We agree with this suggestion and modified text accordingly. We are also glad that our results agree with previous reported literature that has now been correctly cited at lines 351-356 and in the discussion lines 474-476.

18. The discussion reads like paraphrasing the Results section. The manuscript would clearly benefit if the Discussion section had been rewritten short and concisely.

We agree with this suggestion, and we have modified discussion accordingly. We are also willing to shorten the discussion by removing the schematic model that could possibly be used as a graphical abstract.

References

Benechet AP, De Simone G, Di Lucia P, Cilenti F, Barbiera G, Le Bert N, Fumagalli V, Lusito E, Moalli F, Bianchessi V et al. (2019) Dynamics and genomic landscape of CD8(+) T cells undergoing hepatic priming. Nature 574: 200-205

Berg M, Wingender G, Djandji D, Hegenbarth S, Momburg F, Hammerling G, Limmer A, Knolle P (2006) Cross-presentation of antigens from apoptotic tumor cells by liver sinusoidal endothelial cells leads to tumor-specific CD8^+^ T cell tolerance. Eur J Immunol 36: 2960-2970

Boukhaled GM, Harding S, Brooks DG (2021) Opposing Roles of Type I Interferons in Cancer Immunity. Annu Rev Pathol 16: 167-198

Catarinella M, Monestiroli A, Escobar G, Fiocchi A, Tran NL, Aiolfi R, Marra P, Esposito A, Cipriani F, Aldrighetti L et al. (2016) IFNalpha gene/cell therapy curbs colorectal cancer colonization of the liver by acting on the hepatic microenvironment. EMBO Mol Med 8: 155-170

Chambers AF, Groom AC, MacDonald IC (2002) Dissemination and growth of cancer cells in metastatic sites. Nat Rev Cancer 2: 563-572

Cortes E, Lachowski D, Robinson B, Sarper M, Teppo JS, Thorpe SD, Lieberthal TJ, Iwamoto K, Lee DA, Okada-Hatakeyama M et al. (2019) Tamoxifen mechanically reprograms the tumor microenvironment via HIF-1A and reduces cancer cell survival. EMBO Rep 20

Deneve E, Riethdorf S, Ramos J, Nocca D, Coffy A, Daures JP, Maudelonde T, Fabre JM, Pantel K, Alix-Panabieres C (2013) Capture of viable circulating tumor cells in the liver of colorectal cancer patients. Clin Chem 59: 1384-1392

Engstrand J, Stromberg C, Nilsson H, Freedman J, Jonas E (2019) Synchronous and metachronous liver metastases in patients with colorectal cancer-towards a clinically relevant definition. World J Surg Oncol 17: 228

Guidotti LG, Inverso D, Sironi L, Di Lucia P, Fioravanti J, Ganzer L, Fiocchi A, Vacca M, Aiolfi R, Sammicheli S et al. (2015) Immunosurveillance of the liver by intravascular effector CD8(+) T cells. Cell 161: 486-500

Horowitz M, Neeman E, Sharon E, Ben-Eliyahu S (2015) Exploiting the critical perioperative period to improve long-term cancer outcomes. Nature reviews Clinical oncology 12: 213-226

Jeon S, Juhn JH, Han S, Lee J, Hong T, Paek J, Yim DS (2013) Saturable human neopterin response to interferon-α assessed by a pharmacokinetic-pharmacodynamic model. Journal of translational medicine 11: 240

Katlinski KV, Gui J, Katlinskaya YV, Ortiz A, Chakraborty R, Bhattacharya S, Carbone CJ, Beiting DP, Girondo MA, Peck AR et al. (2017) Inactivation of Interferon Receptor Promotes the Establishment of Immune Privileged Tumor Microenvironment. Cancer cell 31: 194-207

Katz SC, Pillarisetty VG, Bleier JI, Shah AB, DeMatteo RP (2004) Liver sinusoidal endothelial cells are insufficient to activate T cells. Journal of immunology 173: 230-235

Kuruppu D, Christophi C, Bertram JF, O'Brien PE (1998) Tamoxifen inhibits colorectal cancer metastases in the liver: a study in a murine model. Journal of gastroenterology and hepatology 13: 521-527

Lalor PF, Shields P, Grant A, Adams DH (2002) Recruitment of lymphocytes to the human liver. Immunol Cell Biol 80: 52-64

Lambert AW, Pattabiraman DR, Weinberg RA (2017) Emerging Biological Principles of Metastasis. Cell 168: 670-691

Pandey E, Nour AS, Harris EN (2020) Prominent Receptors of Liver Sinusoidal Endothelial Cells in Liver Homeostasis and Disease. Front Physiol 11: 873

Sallusto F, Geginat J, Lanzavecchia A (2004) Central memory and effector memory T cell subsets: function, generation, and maintenance. Annu Rev Immunol 22: 745-763

Schreiber G (2017) The molecular basis for differential type I interferon signaling. J Biol Chem 292: 7285-7294

Soares KC, Foley K, Olino K, Leubner A, Mayo SC, Jain A, Jaffee E, Schulick RD, Yoshimura K, Edil B et al. (2014) A preclinical murine model of hepatic metastases. J Vis Exp: 51677

Sorensen KK, Smedsrod, B. (2020) The Liver Sinusoidal Endothelial Cell: Basic Biology and Pathobiology. In: The Liver: Biology and Pathobiology, Sixth Edition pp. 422-434. John Wiley & Sons Ltd.

Stone JD, Chervin AS, Kranz DM (2009) T-cell receptor binding affinities and kinetics: impact on T-cell activity and specificity. Immunology 126: 165-176

Su T, Yang Y, Lai S, Jeong J, Jung Y, McConnell M, Utsumi T, Iwakiri Y (2021) Single-Cell Transcriptomics Reveals Zone-Specific Alterations of Liver Sinusoidal Endothelial Cells in Cirrhosis. Cell Mol Gastroenterol Hepatol 11: 1139-1161

Theeuwes F, Yum SI (1976) Principles of the design and operation of generic osmotic pumps for the delivery of semisolid or liquid drug formulations. Ann Biomed Eng 4: 343353

Van Cutsem E, Cervantes A, Adam R, Sobrero A, Van Krieken JH, Aderka D, Aranda Aguilar E, Bardelli A, Benson A, Bodoky G et al. (2016) ESMO consensus guidelines for the management of patients with metastatic colorectal cancer. Ann Oncol 27: 1386-1422

van Gestel YR, de Hingh IH, van Herk-Sukel MP, van Erning FN, Beerepoot LV, Wijsman JH, Slooter GD, Rutten HJ, Creemers GJ, Lemmens VE (2014) Patterns of metachronous metastases after curative treatment of colorectal cancer. Cancer Epidemiol 38: 448-454

Vidal-Vanaclocha F (2008) The prometastatic microenvironment of the liver. Cancer microenvironment : official journal of the International Cancer Microenvironment Society 1: 113-129

Wang DS, Ohdo S, Koyanagi S, Takane H, Aramaki H, Yukawa E, Higuchi S (2001) Effect of dosing schedule on pharmacokinetics of α interferon and anti-α interferon neutralizing antibody in mice. Antimicrob Agents Chemother 45: 176-180

Wohlfeil SA, Hafele V, Dietsch B, Schledzewski K, Winkler M, Zierow J, Leibing T, Mohammadi MM, Heineke J, Sticht C et al. (2019) Hepatic Endothelial Notch Activation Protects against Liver Metastasis by Regulating Endothelial-Tumor Cell Adhesion Independent of Angiocrine Signaling. Cancer research 79: 598-610

Yu X, Chen L, Liu J, Dai B, Xu G, Shen G, Luo Q, Zhang Z (2019) Immune modulation of liver sinusoidal endothelial cells by melittin nanoparticles suppresses liver metastasis. Nat Commun 10: 574

Zhu Y, Karakhanova S, Huang X, Deng SP, Werner J, Bazhin AV (2014) Influence of interferon-α on the expression of the cancer stem cell markers in pancreatic carcinoma cells. Exp Cell Res 324: 146-156

Ziv Y, Gupta MK, Milsom JW, Vladisavljevic A, Brand M, Fazio VW (1994) The effect of tamoxifen and fenretinimide on human colorectal cancer cell lines in vitro. Anticancer Res 14: 2005-2009